# The transcriptome of HTLV-1-infected primary cells following reactivation reveals changes to host gene expression central to the proviral life cycle

**Aris E. N. Aristodemou**[1], **David S. Rueda**[1,2], **Graham P. Taylor**[1], **Charles R. M. Bangham**[1] *

1 Department of Infectious Disease, Faculty of Medicine, Imperial College London, London, United Kingdom,
2 Single Molecule Imaging Group, MRC-London Institute of Medical Sciences, London, United Kingdom

* c.bangham@imperial.ac.uk

## Abstract

Infections by Human T cell Leukaemia Virus type 1 (HTLV-1) persist for the lifetime of the host by integrating into the genome of CD4+ T cells. Proviral gene expression is essential for proviral survival and the maintenance of the proviral load, through the pro-proliferative changes it induces in infected cells. Despite their role in HTLV-1 infection and a persistent cytotoxic T lymphocyte response raised against the virus, proviral transcripts from the sense-strand are rarely detected in fresh cells extracted from the peripheral blood, and have recently been found to be expressed intermittently by a small subset of cells at a given time. *Ex vivo* culture of infected cells prompts synchronised proviral expression in infected cells from peripheral blood, allowing the study of factors involved in reactivation in primary cells. Here, we used bulk RNA-seq to examine the host transcriptome over six days *in vitro*, following proviral reactivation in primary peripheral CD4+ T cells isolated from subjects with non-malignant HTLV-1 infection. Infected cells displayed a conserved response to reactivation, characterised by discrete stages of gene expression, cell division and subsequently horizontal transmission of the virus. We observed widespread changes in Polycomb gene expression following reactivation, including an increase in PRC2 transcript levels and diverse changes in the expression of PRC1 components. We hypothesize that these transcriptional changes constitute a negative feedback loop that maintains proviral latency by re-deposition of H2AK119ub1 following the end of proviral expression. Using RNAi, we found that certain deubiquitinases, *BAP1*, *USP14* and *OTUD5* each promote proviral transcription. These data demonstrate the detailed trajectory of HTLV-1 proviral reactivation in primary HTLV-1-carrier lymphocytes and the impact on the host cell.

## Author summary

Human T cell Leukaemia Virus type I (HTLV-1) is a retrovirus which causes an aggressive leukaemia or lymphoma, or a chronic inflammatory disease of the central nervous system,

**Data Availability Statement:** Transcript trajectories from this study are made publicly available through a browser-based application

(https://aenaristodemou.shinyapps.io/htlv_reactivation_trajectories/). Image data are made available through the Open Science Framework (https://osf.io/dbsnc/). Sequencing data are deposited in the NCBI's Gene Expression Omnibus (Edgar, 2002), with the accession number GSE234450 (https://www.ncbi.nlm.nih.gov/geo/query/acc.cgi?acc=GSE234450). Remaining data are made available as Supporting information files.

**Funding:** A.E.N.A. is funded by a Wellcome Trust UK 4-year PhD studentship (102126). C.R.M.B. is funded by a Wellcome Trust UK Investigator Award (207477). D.S.R. is supported by a core grant of the MRC-London Institute of Medical Sciences (UKRI MC-A658-5TY10) and a Wellcome Trust Collaborative Grant (206292/Z/17/Z). G.P.T. is supported by the NIHR Imperial Biomedical Research Centre. The funders had no role in study design, data collection and analysis, decision to publish, or preparation of the manuscript.

**Competing interests:** The authors have declared that no competing interests exist.

in a subset ($\approx 10\%$) of affected carriers. Whilst the virus is only intermittently expressed by infected cells, the virus persists in its host by increasing the proliferation rate and survival of infected cells. It is therefore imperative to understand the mechanisms that control the activation and deactivation of the virus. We examined the expression of host and viral genes during HTLV-1 reactivation in cells freshly isolated from patients' blood. The infected cells displayed consistent changes in gene expression over six days. We observed changes in the expression of the Polycomb group of epigenetic modifiers, known to impact HTLV-1 transcription, which may form a negative-feedback mechanism allowing the virus to return to a latent (quiescent) state following activation. We additionally identified a set of three deubiquitinases that increase expression of the virus. Our data detail the changes in gene expression underlying core aspects of the HTLV-1 life cycle in primary cells, and provide a resource for further investigation.

## Introduction

As obligate genomic parasites, retroviruses depend extensively upon host cell machinery to progress through their life cycle. Human T cell leukaemia virus type 1 (HTLV-1) is the first retrovirus discovered infecting humans [1] and is the causative agent of a debilitating neuropathy, HTLV Associated Myelopathy/Tropical Spastic Paraparesis (HAM/TSP), and an aggressive malignancy, Adult T cell Leukaemia (ATL), each affecting $\approx 5\%$ of carriers [2–4]. The risk of disease is correlated with the maintenance of a high proviral load (PVL) in carriers [5–7]. A persistently active cytotoxic T lymphocyte response to antigen produced from the proviral sense-strand accompanies infection, irrespective of disease status [8], and despite the low frequency of expression of viral proteins in freshly isolated cells *in vivo* [9]. The proviral antisense product, *HBZ*, is more readily detected at the RNA level [10–12]; however, it is poorly immunogenic [13,14].

*Ex vivo* culture of HTLV-1-carrier primary peripheral blood mononuclear cells (PBMCs) stimulates proviral sense-strand expression [15,16], dominated by *tax/rex* transcription and Tax protein expression [17]. The gradual accumulation of Rex results in increased nuclear export of unspliced and singly-spliced mRNA species, with consequent depletion of *tax/rex* mRNA and production of structural and enzymatic viral components necessary for virion formation and horizontal transmission. Viral expression is accompanied by changes in host gene expression driven mainly by the pluripotent effects of Tax and HBZ upon host signalling pathways [18], the cell cycle [19] and apoptosis [20,21].

Investigations of HTLV-1-infected samples at the single-cell level have uncovered high levels of heterogeneity in proviral gene expression, suggesting transcriptional bursting of the proviral promoter and intermittent expression [10,11]. Live imaging of Tax expression using a short-lived GFP reporter revealed these transient expression episodes in MT-1 cells [20], a finding later confirmed by the use of a fluorescent timer reporter for Tax expression which allows identification of cells which have recently terminated Tax expression [22]. Intermittent expression offers a plausible mechanism for limiting the exposure of antigen to adaptive immunity whilst reaping the benefits of proviral gene expression, particularly those conferred by Tax and HBZ, which together contribute to infected cell survival and proliferation [18,23].

Integrated into the host genome, the provirus is subject to the molecular mechanisms that regulate host gene expression. Chromatin immunoprecipitation (ChIP) experiments have identified the distribution of multiple epigenetic modifications along the provirus [24–29]. Histone tail post-translational modifications commonly associated with increased

transcriptional activity—H3K27ac, H3K9ac and H3K4me3—are found on the provirus, and differ between Tax[+] and Tax[-] cells [28]. Repressive histone modifications are also associated with the provirus, in particular H2AK119ub1 and H3K27me3 [26,27], respectively deposited by the Polycomb repressive complexes 1 and 2 (PRC1/2) [30]. Proviral reactivation has been demonstrated to depend upon the removal of H2AK119ub1 from the proviral promoter in the 5′ LTR [26], but the roles of epigenetic chromatin modifications in active silencing of the provirus are unknown.

Whilst it is established that HTLV-1 gene expression perturbs host transcription, and many such changes have been invoked in the oncogenesis of ATL, few of these results have been obtained or confirmed in PBMCs from HTLV-1 carriers, and even fewer have examined expression over a longer time frame allowing the observation of changes which occur over both the early and late phases of proviral reactivation. We analysed the transcriptomes of primary HTLV-1-infected CD4[+] T cells cultured *ex vivo* over six days in six subjects with non-malignant HTLV-1 infection and three uninfected control subjects. The resulting RNA-seq timecourse captured the switch from early to late sense-strand gene expression. We observed a marked difference in host gene expression between infected and uninfected samples, with the former showing conserved changes in gene expression over time as cells enter the cell cycle and commence the horizontal transmission of virions. We investigated the pathways stimulated during *ex vivo* culture in uninfected cells, finding that these are already active in non-malignant infected cells at the point of entry into culture. Finally, we observed widespread dysregulation of Polycomb gene expression in infected cells during proviral reactivation.

## Results

### Isolation of HTLV-1-infected population using CADM1

To obtain a profile of HTLV-1 and host gene expression over the reactivation cycle, we used frozen PBMC samples collected from consenting HTLV-1 positive and uninfected volunteers. Our sample consisted of two asymptomatic HTLV-1 carriers (HES and HQL) and four individuals diagnosed with HAM/TSP (TCL, TCQ, TDK and TDX). To obtain sufficient numbers of cells for transcriptome sequencing at multiple timepoints, we selected samples from individuals with a high proviral load ($\geq$ 10%). To reduce bias introduced by abnormally expanded (potentially leukaemic) clones, an oligoclonality index < 0.77 was used as the cutoff for inclusion [31]. The cells were thawed, enriched for CD4[+] lymphocytes by negative selection, sorted for viability and high CADM1 expression and samples were withdrawn for analysis at 0, 24, 48, 96, 120 and 144 hours after the start of *in vitro* culture. RNA was extracted at each timepoint and used for qRT-PCR quantification of HTLV-1 transcripts and quality control prior to submission for sequencing (Fig 1A).

CADM1[high] cell sorting was performed before culture to ensure that infected cell gene expression signatures were not diluted by the presence of uninfected PBMCs. High levels of CADM1 are expressed exclusively by HTLV-1-infected CD4[+] T cells upon their extraction from the circulation (Fig 1C), and represent $\approx$ 65% of the total PVL [32]. Samples from uninfected individuals were processed identically, to control for the effects of thawing, cell-sorting and subsequent *ex vivo* culture. DNA was extracted from a subset of cells following sorting to measure PVL by ddPCR [33]. PVLs in sorted HTLV-1 patient samples were found to be $\approx$ 100% (Fig 1D). Given that each infected cell contains on average a single proviral copy [34], we inferred that our sorted samples consisted entirely of infected cells. No proviral copies were detected in the uninfected control samples (V1, V3 and V4). No proviral copies were detected by ddPCR in the sample for one infected subject, HES, however, this is likely due to

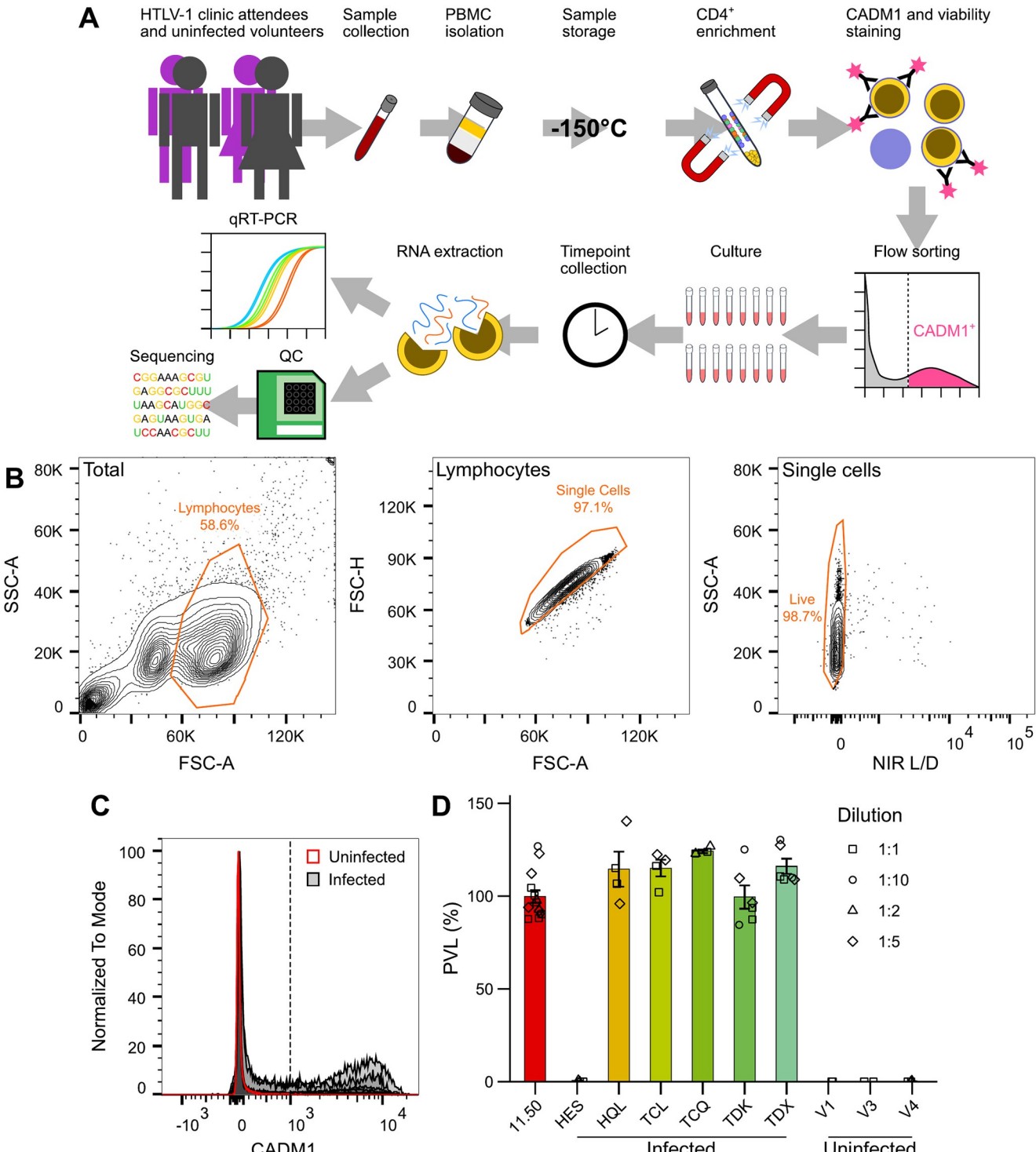

**Fig 1. Sampling of HTLV-1 infected cells along expression cycle *ex vivo*.** (A) Overview of sample preparation and collection. (B) Gating scheme used to obtain single live lymphocytes from peripheral blood lymphocyte samples. (C) Gating of CADM1high cells was performed based on CADM1 levels in uninfected controls. Dotted line represents CADM1 intensity threshold for infected samples. (D) ddPCR measurement of PVL of CADM1high-sorted cells used for RNA-seq at $t_0$. Infected T cell clone 11.50, which contains a single copy HTLV-1 provirus per cell [34,112], was used as a positive control. Columns and error bars represent mean ±SEM.

polymorphism in the primer—or probe—binding sites in the provirus; proviral transcripts were later detected in this sample both by qRT-PCR and RNA-seq (Fig 2).

## Polyclonal infected cell populations respond homogeneously to *ex vivo* culture

Prior to sequencing, RNA from patient samples was subjected to qRT-PCR to examine the trajectories of HTLV-1 transcripts (S1 Fig). A decrease in *tax/rex* transcript abundance was observed following the 24-hour timepoint, together with a decrease in total sense-strand transcripts, detected using a probe complementary to the pX region that is present in all sense-strand transcripts. Signal from the *gag-pro-pol* polytranscript first sense-strand intron, detected using probes complementary to *gag*, remained relatively constant following the first 24 hours (S1 Fig). The overshoot in spliced transcript abundance combined with the monotonic increase of unspliced product is consistent with the precursor-depletion-mediated negative feedback, mediated by Rex in HTLV-1 [17] and by Rev in HIV-1 [35]. A similar overshooting trajectory can be observed in sequenced reads assigned to proviral sense-strand transcript (Fig 2A). The abundance of reads aligning to the proviral sense strand correlated well with the qRT-PCR results obtained using the pX primer pair (Fig 2B). Reads aligning to the HTLV-1 provirus were absent from the uninfected samples (Fig 2A), except in sample V4 $t_{24}$, where a small ($\approx$0.5% of infected sample mean) number of reads aligned to the sense and antisense proviral strands (S2 Fig). These reads most likely represent contamination from infected samples, and this sample was therefore excluded from further analyses. No reads mapping to HTLV-1 were detected in other timepoints for the same patient, suggesting that $t_{24}$ was the only timepoint affected by contamination.

We next examined the ratio of early to late viral transcripts over time, by quantifying the number of reads aligned by STAR [36] to regions corresponding to the Gag CDS or the final Tax exon. We observed a U-shaped trajectory in the ratio of double spliced to unspliced transcripts (Fig 2C), corresponding to an initial switch towards the expression of genes for virion production ($\leq$ 96 hrs.), followed by cells switching back to the expression of spliced transcripts during a second wave of reactivation, either in the same cells or newly reactivating cells. We also examined the levels of antisense transcripts over time, which correlated positively and significantly with sense-strand transcripts (S3 Fig), corroborating previous work noting similar trajectories between transcripts from the two proviral strands [17]. This correlation complicates the assignment of the observed effects on host cell transcription to products of either proviral strand.

Principal component analysis (PCA) was performed on variance-stabilising transformation (VST) normalised transcript count matrices obtained following Kallisto pseudoalignment [37], to visualise relationships in overall gene expression between samples. The first two PCs explain just under 50% of the total variance, with the contribution of subsequent individual PCs tailing off rapidly (Fig 2D). Examination of sample clustering along these two PCs suggested that the main sources of variance in the data were HTLV-1 infection status and time in culture. The difference in mean age between the infected and uninfected subjects might contribute to this variance (S4 Fig). Infected samples showed strong within-timepoint clustering up to and including 96 hours, suggesting a common gene expression progression profile. In contrast, uninfected samples changed relatively little following the initial stimulus of their introduction into culture (Fig 2E).

An additional comparison was performed with the transcriptome obtained by Kiik et al. [22], which utilised a fluorescent timer protein to distinguish between the successive phases of Tax expression in clonal infected T cells (S5 Fig). Samples from Kiik et al. [22] clustered

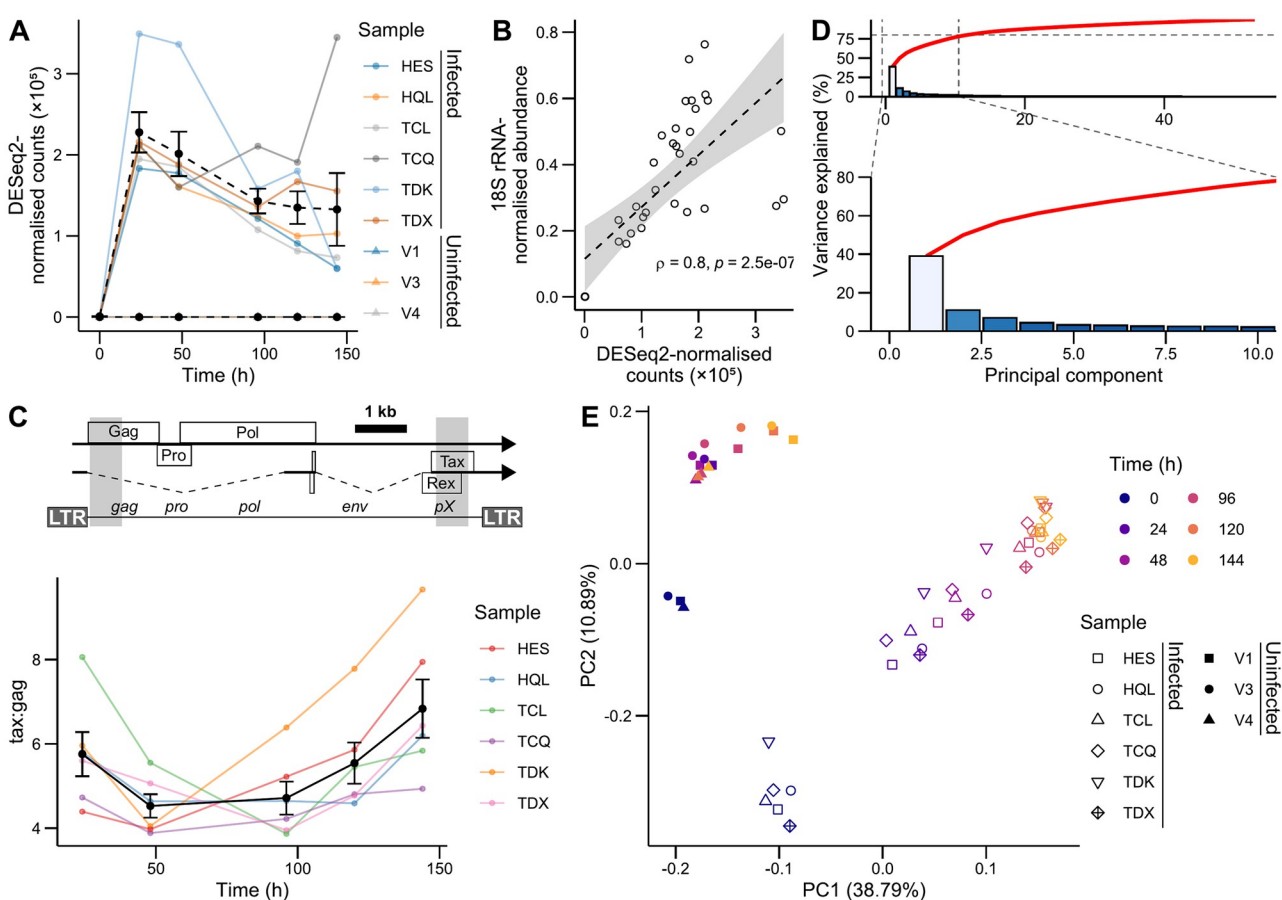

**Fig 2. RNA-seq dataset captures progression of infected cells along HTLV-1 expression stages.** (A) Trajectory of sequenced reads mapped to the HTLV-1 sense-strand, excluding the LTRs. (B) Correlation between RNA-seq and qRT-PCR HTLV-1 transcript trajectories. Dotted line represents linear model fit. Shaded area represents linear model 95% confidence interval. Statistics shown were calculated using a paired Spearman rank correlation test. (C) Above: Schematic of proviral sequence with *tax/rex* and *gag-pro-pol* polytranscript shown. Dotted lines represent introns, boxes represent ORFs, and lines represent untranslated regions. Shaded regions correspond to AB513134 regions 804–1,433 and 7,400–8,029, used to quantify reads aligning to *gag* and *tax* regions, respectively. Below: Ratio of reads aligning to proviral *tax* and *gag* regions, corresponding to the ratio of spliced to unspliced transcripts. Summary line and error bars represent mean ±SEM. (D) Scree plot showing contribution of principal components to total variance observed, with red line representing cumulative sum. Zoom inset focuses on principal components accounting for ≈ 80% of variance in the data. (E) PCA bi-plot of infected and uninfected cell gene expression.

together with infected samples in the second principal component. Blue, double-positive and red timer protein populations, representing respectively early, mid- and post-Tax expression cell states, clustered together with the 24-hour samples. The PBMC dataset therefore captures a greater degree of diversity over proviral expression. The 96–144-hour time-points did not coincide with the red population. This observation suggests that Tax expression might not have terminated by 144 hours. Double-negative populations were positioned between the 0 and 24-hour timepoints, potentially reflecting changes related to culture adaptation, as they fall closer to the late-timepoint uninfected samples in the first principal component.

These results indicate that the observed trajectory captures at least part of the switch between early and late proviral transcripts. Within the period of observation of 144 h, infected cells progress through distinct stages of gene expression caused by viral gene expression. This progression is conserved between individual patients, irrespective of disease status.

## Discrete changes to cell state occur following HTLV-1 reactivation

To obtain an overview of the changes in gene expression underlying differences between measurements at different timepoints, we performed differential expression analysis of Kallisto transcript pseudocounts using DESeq2 [38]. An interaction model accounting for differential expression over time between infected and uninfected cells was used, yielding 7,753/28,693 significantly differentially expressed genes (Likelihood-ratio test; Benjamini-Hochberg FDR-adjusted $p < 0.05$). A heatmap of these differentially expressed genes (Fig 3A) showed clustering between samples consistent with that seen in the first two principal components (Fig 2E): uninfected samples clustered separately from infected samples and showed similar expression patterns after 24 hours, with little clustering within individual timepoints. In contrast, infected samples clustered well by timepoint up to 120 hours, at which point patient-specific differences became dominant (Fig 3A).

We next examined further the differences between infected timepoints, to identify processes altered following HTLV-1 reactivation in infected cells. We performed over-representation analysis (ORA), using clusterProfiler [39] on gene sets identified as differentially expressed between pairs of successive timepoints in infected cells. Over-represented Gene Ontology (GO) biological process terms [40,41] within levels 5–10 of the GO term hierarchy were kept and further filtered to reduce redundancy, retaining only the lowest FDR-corrected $p$-value of terms with $\leq 50\%$ overlap (Fig 3B).

The 0–24-hour interval was characterised by differential expression of genes annotated as being involved in the response to viral gene expression. However, these genes likely reflect a general response to stress, as their trajectory is shared with the uninfected control samples (Fig 3C). Also over-represented in this interval were genes involved in ribosome biogenesis, ribonucleoprotein complex organisation and RNA catabolism, likely reflecting the initial adaptation to the novel *in vitro* nutritional environment. Viral gene expression is more likely to exert an additional impact on genes associated with these nutrient stress-related GO terms, as their trajectories can be seen to vary from those in uninfected controls responding to *ex vivo* culture alone (Fig 3C).

DNA replication was triggered early in infected samples, and multiple related terms were enriched in the subsequent 24–48 and 48–96 hour intervals (Fig 3B). We examined the trajectories of transcripts known to peak at distinct cell cycle stages, obtained from Cyclebase 3.0 [42]. An increase in transcripts associated with all cell cycle stages was observed following $t_{24}$, indicating asynchronous entry into the cell cycle, given the simultaneous upregulation of all stage markers (S6 Fig). It is clear that this inferred entry into the cell cycle is due to the expression of proviral genes, as the trajectories of cell cycle associated terms in uninfected samples are distinct from those in infected samples (Fig 3C and S6 Fig).

Terms associated with transition into the final two timepoints were more heterogeneous, comprising genes involved in synapse organisation and vesicle formation, as well as ion and metabolite transport. Terms related to vesicle formation or transport, and synapse-related terms, may represent horizontal infection, which occurs across a cell-cell junction and may involve vesicle-based transfer of virions [43,44]. Horizontal infection is a key outcome of proviral plus-strand expression. At least some HTLV-1-infected cells must preserve the potential to reactivate, in order to retain the ability to infect a new host. Horizontal transmission can occur during *ex vivo* culture of CD4$^+$ patient cells [43]. The transfer of new virions between cells was evidenced by proviral genome copy numbers, measured by ddPCR of extracted DNA, which surpassed one copy per cell (S7A and S7B Fig). The number of reverse-transcribed proviral genomes grew exponentially, the increase first becoming apparent in the 24–48 hour interval (S7A Fig). In contrast, synapse- and vesicle formation-related terms were

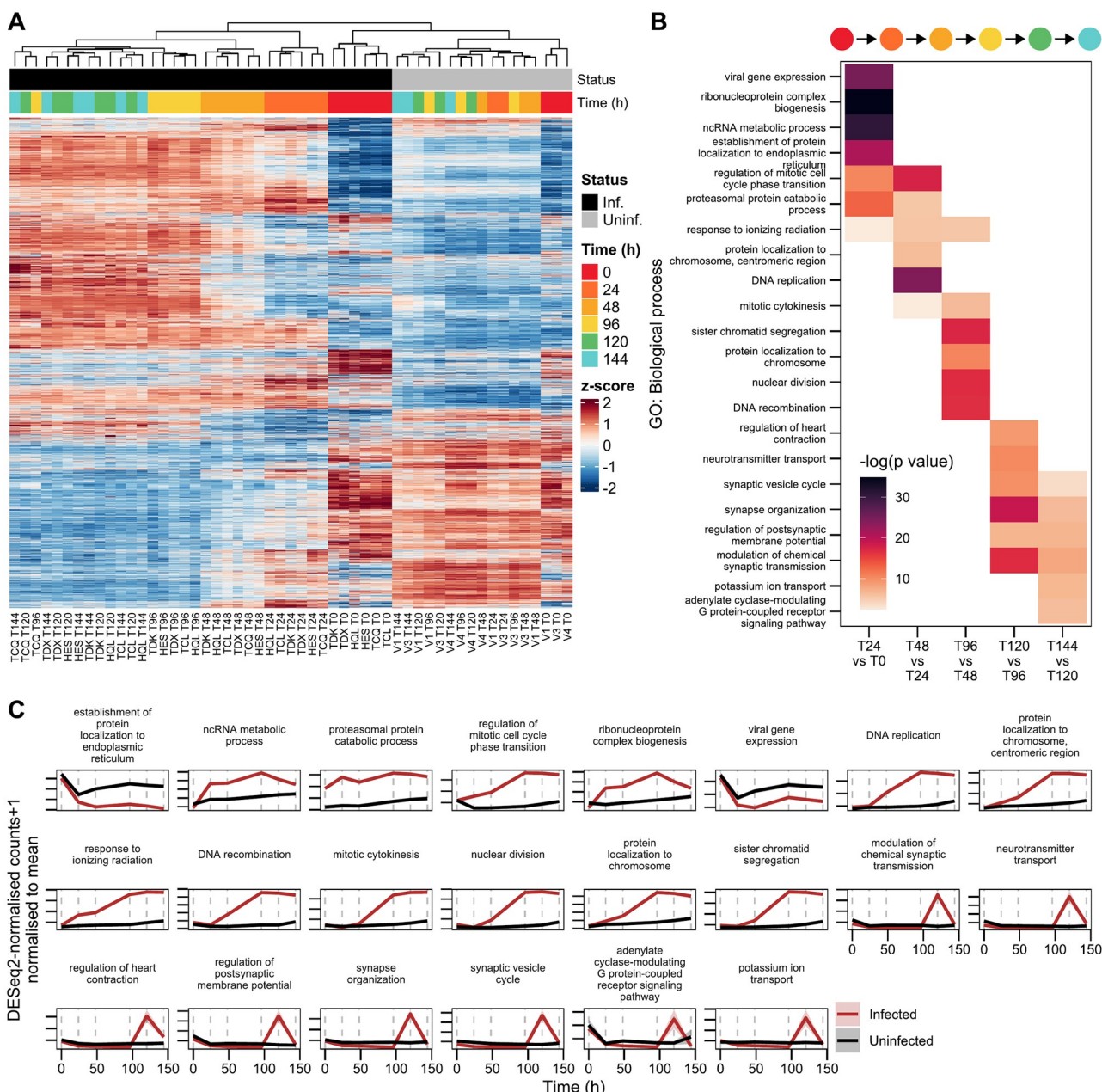

**Fig 3. Timepoints separated by distinct changes in gene expression specific to infected cells.** (A) Heatmap of genes differentially expressed over time between infected and uninfected samples. (B) Top five (based on FDR-corrected *p* value) non-synonymous gene ontology (GO) biological process terms associated with each interval. (C) Summary of mean-normalised trajectories of genes for GO terms presented in panel B. Discontinuous vertical lines indicate sampling timepoints. Shaded areas represent ±SEM.

enriched exclusively the in the 96–120 and 120–144 hour intervals (Fig 3B and 3C). The observed rate of these increases in reverse-transcribed proviral genomes indicates that these processes are not involved in horizontal transmission. Instead, they may be associated with the action of accessory proteins thought to affect cellular ion concentrations, for example p12 [45] or p13 [46,47].

Imaging of proviral transcripts using single-molecule RNA FISH revealed clustering of signal from both introns and exons of the provirus, consistent with accumulation of whole virus genomes (S7C Fig). We quantified the fraction of imaged cells showing evidence of unspliced RNA clusters near the cytoplasmic periphery or between cells (S7C Fig) over time in both patient samples. Since the cells were mixed before mounting on coverslips, it is likely that many cells showing polarisation of viral RNA had lost contact with other cells. Therefore, cells with clustered RNA not in contact with other cells were also considered. The fraction of cells showing these RNA clusters increased over time (S7D Fig), the trajectory resembling the observed increase in reverse-transcribed proviral genome counts (S7A Fig), further suggesting the horizontal transmission of virions in culture.

We then obtained a list of genes associated with horizontal transmission, compiled by Gross and Thoma-Kress [48], and examined their trajectories in infected and uninfected samples (S7E Fig). In light of the timing of the PVL increase observed in reactivating cells (S7A Fig), it is likely that terms which showed increased expression late during reactivation, such as *FSCN-1*, *VCAM-1*, *SDC-1*, *CCL22* and collagen had little impact upon the infection process in *ex vivo* patient samples; their upregulation was not necessary for the transfer of virions into target cells. Conversely, genes that showed substantial changes in expression before the 48-hour timepoint might play a more central role. Additional interesting patterns were observed in the trajectories of individual genes. First, the majority (21/32) of the infection-related gene set tested was significantly differentially expressed in infected cells, suggesting that most related processes are triggered by proviral gene expression, rather than by the stress which reactivates the provirus upon *ex vivo* culture. The interaction between ICAM-1 and LFA-1 is important for establishing contact between an infecting cell and its target [49]. *ICAM-1* and *LFA-1* were regulated in the opposite sense in infected cells. Opposing expression of these genes might limit horizontal transmission between infected cells, so limiting superinfection [34].

## HTLV-1-infected cells and uninfected cells respond similarly to *ex vivo* culture

Multiple pathways influence the degree of proviral reactivation following the stimulation of infected cells; however, without appropriate uninfected controls it is difficult to distinguish between pathways and factors which trigger reactivation and those which merely modulate its intensity. We examined genes significantly differentially expressed in uninfected cells in the 0–24 hour interval (Wald test; BH FDR-corrected $p < 0.05$) to identify pathways affected by the combined stimuli of thawing from -150˚C storage and ex vivo culture responsible for proviral reactivation. ORA was performed using the Hallmark gene sets from the Molecular Signatures Database [50]. Several general stress and nutrient response pathways were highlighted (Fig 4A), reflecting the introduction into the culture environment and thawing from -150˚C storage. c-Myc and mTORC1 signaling are known to coordinate the response to nutrient stress [51,52], with the former being influenced by the latter [53]. The unfolded protein response is similarly triggered by diverse extracellular stimuli or internal changes in metabolite levels and impacts multiple aspects of protein production and secretion [54]. NF-$\kappa$B expression can be activated by a wide variety of stressors [55].

We hypothesised that HTLV-1-infected cells are altered in their ability to react to these stimuli, in order to regulate proviral reactivation. We compared the $log_2$-fold changes in genes associated with these Hallmark terms between 0–24 hours in infected and uninfected samples. Infected cells displayed greater differences in the expression of these genes for all terms examined, aside from protein secretion (Fig 4B). This result is consistent with the known impact of

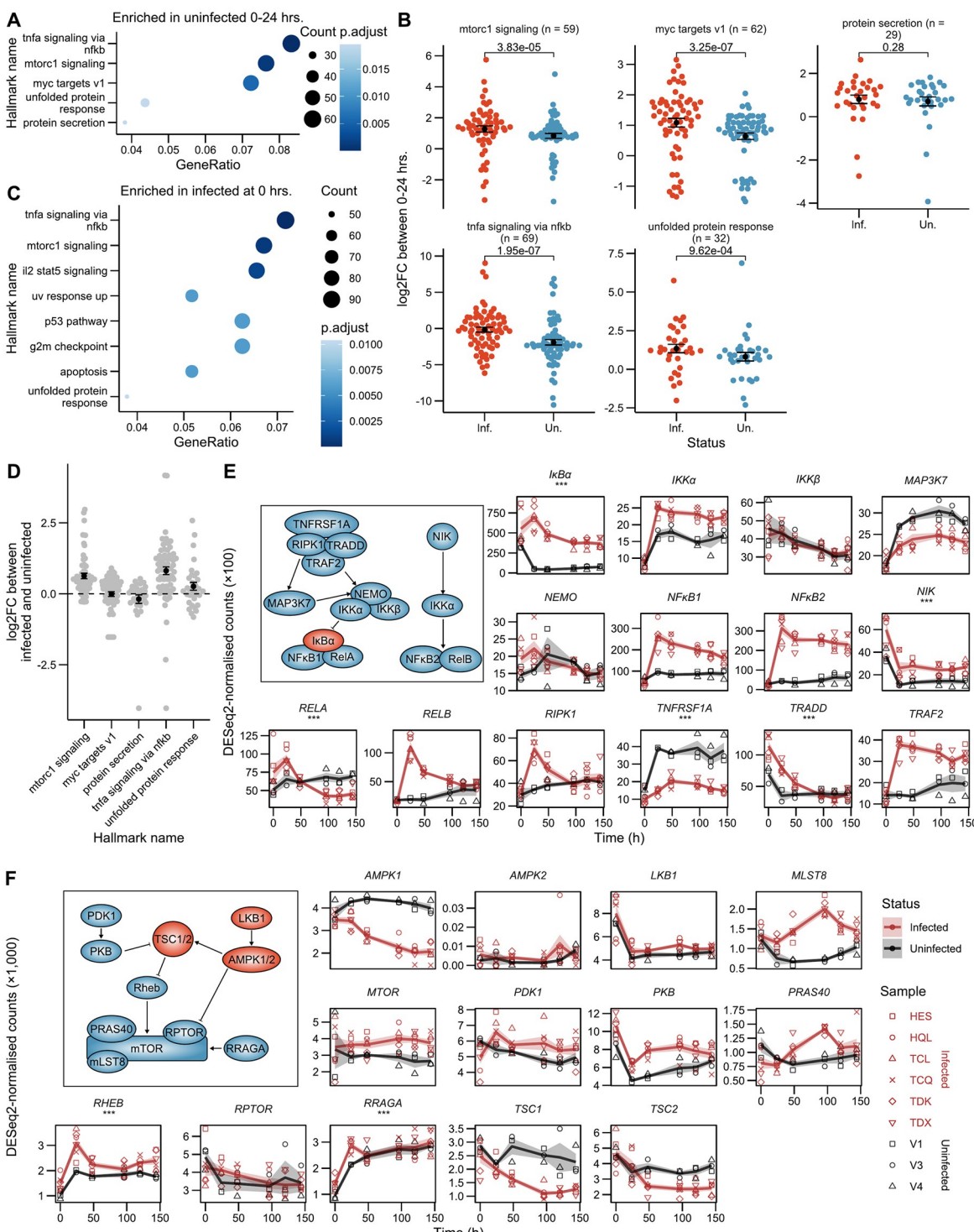

**Fig 4. HTLV-1 infection does not alter responsiveness to reactivating factors.** (A) ORA using Hallmark gene sets of genes differentially expressed in uninfected samples during their introduction to *ex vivo* culture (0–24 hrs.). (B) *log₂*-fold changes during reactivation (0–24 hrs.) in infected and uninfected samples. BH FDR-corrected paired Wilcoxon test *p* values shown. (C) ORA of genes differentially expressed between infected and uninfected samples at 0 hours. (D) Differential expression of genes within Hallmark sets between infected and uninfected samples at 0 hours. Summary bars represent mean ±SEM. (E–F) Trajectories of key TNFα signaling via NF-κB, and mTORC1 pathway components. Asterisks indicate genes which are significantly differentially expressed between infected and uninfected samples at 0 hours (Wald test; BH FDR-corrected *p* < 0.05). Summary lines and shaded areas represent mean ±SEM. Insets highlight factors with positive (blue) or inhibitory (red) effect on pathway.

HTLV-1 products on signalling in each of these pathways [56–59]. To distinguish between potential Tax transactivation and increased sensitivity to these stimuli, we examined the differences between infected and uninfected cells at 0 hours, prior to Tax-mediated transactivation. ORA of genes differentially expressed between infected and uninfected samples at $t_0$ revealed that genes associated with either NF-$\kappa$B, mTORC1 signalling or the unfolded protein response were already differentially expressed before *ex vivo* culture (Fig 4C). Indeed, genes in these pathways which are differentially expressed by uninfected samples in response to stress (Fig 4A), were already upregulated in infected cells at $t_0$ (Fig 4D), relative to uninfected samples.

We then examined the levels of key mediators of the NF-$\kappa$B and mTORC1 pathways to identify potential differences in expression levels which might suggest differential sensitivity to reactivation stimuli, as opposed to the pre-activation of these pathways at $t_0$ (Fig 4E and 4F). Multiple genes were differentially expressed in infected relative to uninfected samples, again highlighting the impact of HTLV-1 expression on these pathways. However, in the NF-$\kappa$B pathway, only 4/11 of these factors were significantly differentially expressed at $t_0$. Moreover, we observed both positive effects on the pathway (upregulation of *TRADD* and *RELA*) and suppressive effects (downregulation of *TNFRSF1A*, upregulation of *I$\kappa$B$\alpha$*), suggesting that infected cells are not differentially sensitive to NF-$\kappa$B stimuli prior to reactivation (Fig 4E). Similarly, factors facilitating mTORC1 signaling did not differ between infected and uninfected cells at $t_0$, with the exception of *RRAGA* and *RHEB*, where increased expression was apparent in the infected cells (Fig 4F). Therefore, it is unlikely that these pathways are more responsive in infected cells. Instead, it is more likely that there is some degree of pre-activation of the NF-$\kappa$B and mTORC1 pathways in infected cells responsible for their differential expression between infected and uninfected cells at $t_0$.

These results indicate that *ex vivo* culture triggers a set of stress-related responses in all samples tested, and HTLV-1-infected cells showed an amplified response to these stimuli. This amplified response is more likely to be the result of synergism with HTLV-1 reactivation than a greater sensitivity to these stimuli, because key mediators of affected pathways were not differentially expressed prior to stimulation.

## Histone ubiquitination factors most closely correlate with sense-strand transcription

We postulated that proviral transcription may trigger the expression of host factors that limit the duration of proviral expression. Kiik et al. [22] observed that the expression of several components of the non-canonical PRC1 complex, which catalyses ubiquination of H2AK119, correlated with sense-strand expression in clonal HTLV-1-infected T cells. We obtained a set of genes correlated with sense-strand transcription by performing Spearman rank correlation tests between the proviral sense-strand and host transcripts, followed by correction for multiple hypothesis testing (Benjamini-Hochberg false discovery rate) [60] (Fig 5A). We then performed an ORA for enriched GO:BP terms containing the term "histone" within their description. No term was significantly over-represented (ORA hypergeometric test; BH FDR-corrected $p < 0.05$), however, the three top ranked terms related to histone ubiquitination (Fig 5B and S1 Table). The genes within these groups are components of PRC1 complexes (*RYBP*, *RING1B/RNF2*, *BCOR*, *KDM2B*) and other ubiquitin ligases (S2 Table). Over 90% of these histone ubiquitination-associated genes were positively correlated with sense-strand expression (Fig 5C). These results are consistent with the notion that PRC1 contributes to the active termination of the HTLV-1 plus-strand burst. In addition, the majority of these genes were differentially expressed between infected and uninfected cells (Fig 5D), indicating an association with proviral expression rather than with the response to stress.

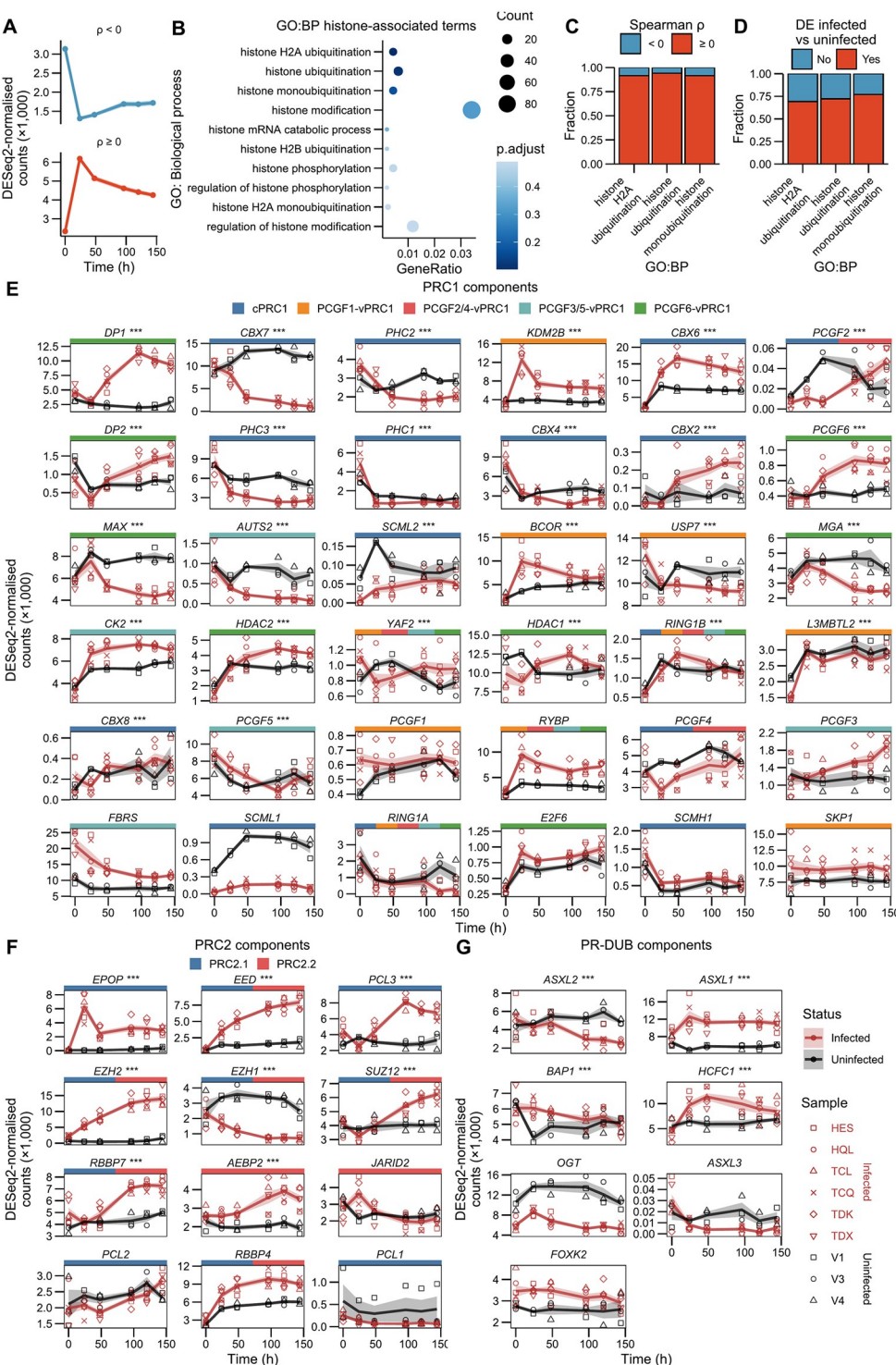

**Fig 5. Polycomb effectors are differentially expressed during reactivation.** (A) Mean trajectories of genes significantly (paired Spearman rank correlation test; FDR-corrected $p < 0.05$) correlated with sense-strand transcript levels. Shading represents ±SEM. (B) Top 10 terms ranked by $p$-value. (C) Fraction of genes annotated as histone ubiquitination-associated correlated either positively or negatively with sense-strand expression. (D) Fraction of genes annotated as histone ubiquitination-associated differentially expressed over time between infected and uninfected samples (Likelihood ratio test; BH FDR-corrected $p < 0.05$). (E) Expression trajectories of PRC1 complex components. Asterisks denote significant differential expression between infected and uninfected samples (Likelihood ratio test; BH FDR-corrected $p < 0.05$). Coloured boxes represent PRC complexes comprising each gene [30]. Shaded areas

represent ±SEM. (F) Trajectories of PRC2 components. (G) Trajectories of PR-DUB complex components and associated factors.

We next examined changes in Polycomb group gene expression induced by proviral reactivation, to identify potential coordination between components of individual complexes which might distinguish a silencing mechanism. Multiple PRC1 complex components were differentially expressed over time between infected samples and uninfected controls (Fig 5E). Most PCGF6-vPCR1 complex components were upregulated over the culture period, again suggesting that PRC1 contributes to silencing following reactivation, although the HTLV-1 provirus possesses few high-quality matches to E- and T-box motifs (S8 Fig) recognised by the MAX or MGA components of PCGF6-vPRC1 at a subset of its genomic target sites [61]. Similarly, PCGF1-vPRC1 components were largely upregulated, potentially guiding the complex to the GC-rich unmethylated regions in the LTRs which are associated with proviral sense-strand expression. Components unique to cPRC1 were mostly downregulated, with the exception of *CBX6* and *CBX2*.

Core PRC2 components (*EED*, *EZH2*, *SUZ12* and *RBBP4/7*) increased over time in culture (Fig 5F), potentially affecting the maintenance of the H3K27me3 histone mark observed during reactivation [27]. Contrary to the other core components, *EZH1* was downregulated. Of the PRC2-variant subcomponents, *EPOP* showed the largest change, with other subcomponents changing relatively little, remaining expressed at some level throughout. PR-DUB deubiquitination complex components were not greatly altered during reactivation: with the exception of *HCFC1*, all were expressed in cells at the point of culture (Fig 5G). As expected, *ASXL3* levels were expressed at a very low level throughout (Fig 5G), reflecting their specific expression in neural tissues [62].

## Diverse deubiquitinases are involved in HTLV-1 reactivation

H2AK119ub1 levels at the promoter were found to influence proviral reactivation through the use of a broad-specificity deubiquitinase (DUB) inhibitor, PR-619 [26]. PR-619 has multiple targets, hence the precise DUBs involved in HTLV-1 reactivation are not known. We examined our dataset for evidence of known PR-619 target DUBs being significantly upregulated during reactivation, i.e. the $t_0$–$t_{24}$ interval. A PR-619 target list consisting of 27 genes was obtained from the manufacturer's website (https://www.selleckchem.com/products/pr-619.html). Of these PR-619 target genes, 13 were found to be significantly upregulated in infected cells between $t_0$ and $t_{24}$ (Fig 6A).

We selected three DUBs for knockdown using siRNA-induced RNA interference, *USP16*, *USP14* and *OTUD5*. These DUBs were selected due to their high fold change during reactivation (*USP14*), or a stark difference in expression trajectory between infected and uninfected cells (*OTUD5*) or published involvement in H2A deubiquitination (*USP16*) [63,64]. We also targeted *BAP1*, which was not significantly differentially regulated during reactivation (Fig 6A), but is targeted by PR-619 and is the deubiquitinase present in the PR-DUB complex. RNAi of genes involved in reactivation is impracticable using the PBMC reactivation system, because the time needed to deplete transcript and protein levels exceeds the reactivation timeframe. We therefore incubated three clonal infected T cell lines established from naturally infected cells [34] with siRNA targeting *USP14*, *USP16*, *OTUD5* or *BAP1*. These cells are maintained continuously in culture and are therefore asynchronous with respect to proviral expression [10].

RNAi reduced target transcript levels, as measured at 72- and 96-hours post siRNA introduction, with mean knockdown efficiency as high as 84.8%, relative to the non-targeting

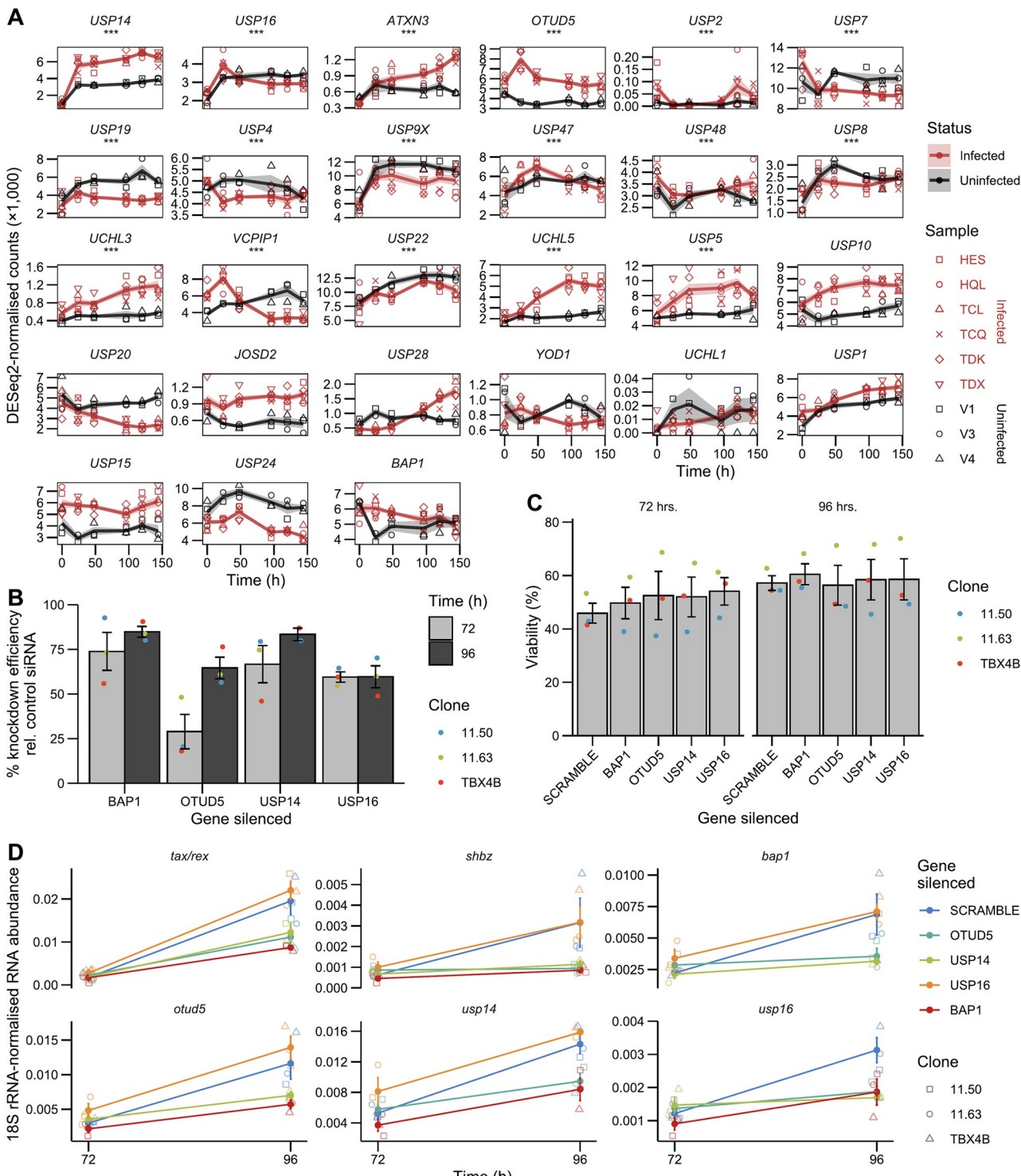

**Fig 6. Knockdown of BAP1, OTUD5 or USP14 reduces proviral transcript levels in infected cells.** (A) Trajectories of PR-619 targets in infected and uninfected cells. Asterisks denote genes significantly differentially expressed in infected samples between 0–24 hours (Wald test; BH FDR-adjusted $p < 0.05$). Summary lines and shaded area represent mean ±SEM. (B) Knockdown efficiency following incubation with siRNA. Dot colours represent individual infected T cell clones carrying distinct proviral integration sites. Error bars ±SEM. (C) Viability measurements at 72 and 96 hours following the addition of siRNA targeting DUBs. Error bars ±SEM. (D) Changes in transcript levels upon DUB inhibition after 72 and 96 hours. Error bars ±SEM.

siRNA control (Fig 6B). The siRNA delivery protocol reduced cell viability; however, individual deubiquitinase knockdown did not appear to reduce viability further. The impact of the siRNA protocol on cell viability is likely due to the omission of fetal calf serum (FCS) during incubation with siRNA, since the viability increased within 24 hours of the reintroduction of FCS to the culture medium (Fig 6C). Knockdown of *BAP1*, *OTUD5* or *USP14* resulted in the reduction of *tax/rex* transcript levels, suggesting a role for these genes in reactivation of sense-strand transcription (Fig 6D). This reduction in plus-strand mRNA abundance was accompanied by similar changes in *shbz* transcript levels.

Knockdown of any one of these three DUBs resulted in a reduction in the levels of the other two, although the siRNA sequences used were specific to each respective protein (S9 Fig). Knockdown of *USP16*, which is known to be involved in H2AK119ub1 deubiquitination, appeared to have no effect on any of the transcripts examined, although it itself is influenced by both *BAP1*, *OTUD5* and *USP14*. This lack of effect might be due to incomplete depletion, as knockdown efficiency only reached 60%. Although the knockdown efficiency was similar between 72 and 96 hours of siRNA incubation, a reduction in proviral transcripts or non-target DUBs only became apparent at 96 h (Fig 6D). FCS was reintroduced at 72 hours, following $t_{72}$ RNA sample collection, according to the manufacturer's instructions. It is likely that FCS addition triggered synchronous reactivation of some cells, which then made the effects of DUB depletion evident, although protein levels might have been effectively depleted only at 96 hours. Regardless, these results confirm previous findings regarding the importance of deubiquitination for HTLV-1 reactivation and implicate specific deubiquitinases in the process.

## Discussion

HTLV-1 is persistently expressed at the cell population level *in vivo*, as evidenced by the persistent activation of cytotoxic T lymphocytes against HTLV-1 in infected individuals [8]. However, only a small fraction of infected cells in peripheral blood express sense-strand transcripts at a given time [9,11]. Proviral expression can be induced by *ex vivo* culture [15,16,65]. Here, we have examined host transcriptome changes over six days in CD4+ T cells from individuals with polyclonal, nonmalignant HTLV-1 infections, comparing it to the behaviour of uninfected CD4+ cells presented with identical stimuli.

Infected cells displayed a distinct, temporally conserved response to *ex vivo* culture. The processes distinguishing these timepoints are key to proviral survival, and include the modification of host cell signalling pathways, the initiation of horizontal infection, and entry into the cell cycle. Infected cells showed changes in gene expression consistent with cell cycle entry following their introduction to *ex vivo* culture. In contrast, uninfected samples did not undergo the same changes, as evidenced by their trajectories of cell-cycle related terms (S6 Fig). The differential response to the culture and thawing stimuli provides further support for the maintenance of PVL through increased proliferation, driven by proviral gene expression [66–69]. The timing and ultimate outcome of cell cycle entry might vary substantially between infected cells, given the observed simultaneous upregulation of markers of distinct cell cycle stages. Indeed, it has been suggested that the intensity of sense-strand expression determines the fate of the HTLV-1-infected cell, with low expression levels resulting in proliferation and intense expression resulting in senescence and death [70].

We observed an increase in the copy number of viral DNA per cell in DNA extracted from patient PBMCs cultured over time, corroborating prior observations of horizontal transmission in patient PBMCs cultured *ex vivo* [43]. This increase likely reflects the reverse transcription of vRNA following horizontal transmission of virions into target cells. Examination of several genes thought to be involved in the horizontal infection process confirmed their

differential expression in infected samples. The expression trajectories of these genes varied substantially, with some becoming expressed only following the observed increase in reverse transcribed proviral copies. One notable example is *VCAM-1*, which is nearly undetectable at 0 hours and increases appreciably only by $\approx 48$ hours. Interestingly, antibody-mediated blocking of VCAM-1 reportedly had little effect on HTLV-1 transmission through EVs, relative to ICAM-1 [44]. A similar pattern was observed for *FSCN-1*, which has been shown to influence the efficacy of infection *in vitro* between the ATL cell line MT-2 and Jurkat cells [48]. These observations highlight important differences between primary cells and the models used in earlier work, supporting increased use of primary cells wherever possible. We also observed opposite effects of HTLV-1 expression on *ICAM-1* and *LFA-1*, which interact between infected and uninfected cells to promote virological synapse formation [49,71]. Despite recent reports of possible recombination events in the evolutionary history of some HTLV-1 strains [72,73], the presence of a single integrated provirus in the majority of infected cells [34] suggests HTLV-1 has mechanisms to restrict superinfection [74]. The downregulation of LFA-1 during reactivation might reduce the likelihood of superinfection following an encounter between infected cells, although LFA-1 is expressed in infected cells before reactivation at a higher level than in uninfected cells.

Infected cells showed a greater response to stimuli associated with thawing and *ex vivo* culture than uninfected controls. Since the key regulators of the pathways activated by these stressors were not differentially expressed at $t_0$, we concluded that this greater response was due to the effects of proviral gene expression, which is known to induce both NF-$\kappa$B [75,76] and mTOR signaling [59,77,78], as opposed to an altered responsiveness to external stimuli. Nevertheless, it was evident that infected cells displayed some activation of these pathways at $t_0$, relative to uninfected cells, suggesting chronic activation of these pathways in nonmalignant infected cells.

The proposed mechanism for mTOR activation by HBZ [78] is compatible with activation prior to reactivation, as HBZ is detectable immediately upon isolation of infected cells from patient blood [11,12]. How NF-$\kappa$B activation is maintained in the absence of constant sense-strand expression is less clear. HBZ expression is thought to antagonise NF-$\kappa$B activity, through its suppressive effects on RelA [79]. In addition, the effects of Tax expression on NF-$\kappa$B have been reported as transient and correlated with the expression of Tax [22]. Indeed, we observed trajectories similar to sense-strand expression for multiple mediators of NF-$\kappa$B signaling, so it is unclear how transient proviral expression could account for constitutive activation. In ATL cells, NF-$\kappa$B is chronically activated [80] and mutations affecting the NF-$\kappa$B pathway are frequently detected [81]. Oncogenic driver mutations can be detected in clonally expanded premalignant infected cells up to 10 years before the onset of ATL [82]. However, the infections examined here are nonmalignant and polyclonal: somatic mutations are therefore unlikely to account for the chronic activation of NF-$\kappa$B signalling in these cell populations.

Before the initiation of *ex vivo* culture and the experimental start point ($t_0$), cells were extracted from donors, isolated using gradient density centrifugation, frozen, thawed, stained, and sorted: this procedure imposes stress on the cell, which might therefore already start to activate proviral transcription. Timecourse analyses of the dynamics of reactivation of immediate-early stress response genes show that *c-FOS* transcript levels reach a maximum at $\approx 15$ minutes post-stimulation, followed by a return to basal levels by 60 minutes [83]. *c-FOS* is expressed at a high level at the 0-hour timepoint in our dataset (S10 Fig), and is then near zero at all subsequent timepoints. This suggests that experimental $t_0$ lies at under 60 minutes following the initial stimulus. The difference in mean age between our infected and uninfected controls might also contribute to our result. Ageing has been observed to influence the steady-

state expression of stress-pathways, including both mTORC1 [84] and NF-$\kappa$B [85]. It is possible that age-related upregulation of these pathways contributes to the increased pathogenesis of HTLV-1 in older individuals.

Proviral sense-strand expression displayed an overshoot, which was also apparent in qRT-PCR using primers against either *tax/rex* or the pX region. This pattern is apparent in several publications, at both the RNA and protein levels [17,24,25]. The overshoot shape can arise through negative feedback, and does so through precursor transcript depletion in HIV, where Rev removes full-length proviral transcripts before splicing can occur [35]. A similar mechanism likely exists in HTLV-1, where the RNA export factor Rex can exert negative feedback upon sense-strand expression [17,86]. This negative feedback has been proposed to lead to oscillations in Tax expression [86], consistent with live imaging of proviral expression using a d2EGFP reporter [20].

Host gene expression, potentially driven by the provirus, may also contribute to the induction or maintenance of a latent state following reactivation. Histone modification levels influence proviral transcription [24,26], are associated with Tax expression [28] and change during reactivation [26,27]. Genes associated with histone ubiquitination, the removal of which is necessary for proviral reactivation [26], were found to be better represented amongst genes which correlate with sense-strand expression than other histone modifications, indicating a more direct relationship with proviral expression. These findings are consistent with our previous results using naturally-infected T cell clones [22].

We found that HTLV-1 reactivation alters the expression of multiple Polycomb group genes, with several PRC1 components showing trajectories which mirror HTLV-1 expression, whilst core PRC2 components show a general increase over time. Microarray-derived gene expression patterns observed in ATL samples relative to healthy CD4$^+$ T cells have previously shown that multiple PRC2 and a subset of PRC1 components were upregulated in infected samples [87,88]. PRC2 upregulation may be related to the increased EZH2 occupancy detected at the proviral promoter post-reactivation [26]. H3K27me3 (deposited by PRC2) levels are also maintained through reactivation [27]. It is possible that the maintenance of H3K27me3 by PRC2 during reactivation allows the rapid re-establishment of H2AK119ub1 by cPRC1, in accordance with the canonical model of sequential Polycomb repression [30]. Transient induction of immediate-early genes by vascular endothelial cell growth factor in human umbilical vein endothelial cells occurs through the activation of their bivalent (H3K4me3 and H3K27me3 positive [89]) promoters whilst H3K27me3 is maintained and EZH2 accumulates at the promoter. cPRC1 then deposits H2AK119ub1, coinciding with the reduction in gene expression [90]. The sense-strand promoter shares several functional features with immediate-early genes, including bivalent promoter histone marks [25,27,28], independence from *de novo* protein synthesis for reactivation [26] and responsiveness to p38-MAPK signaling [26]. Whilst it is tempting to draw parallels between these systems, the immediate-early gene response studied by [90] is far more short-lived ($< 4$ hrs.), with transcript levels falling by 60 minutes post-stimulation. It is possible that Tax-mediated transactivation postpones this decrease, thereby maintaining sense-strand expression.

Transcript trajectories alone are insufficient to infer a mechanism of epigenetic regulation. It will be important to investigate the kinetics of H2AK119ub1 re-establishment at the provirus, and its potential role in active silencing of the provirus. Regardless, the findings of [26] concerning the impact of DUB function on proviral transcription have been confirmed by the DUB RNAi results presented. We found that siRNA knockdown of *USP14*, *OTUD5* or *BAP1* reduced *tax/rex* transcript counts in clonal infected T cells. The importance of H2AK119 deubiquitination to sense-strand transcription has not previously been assayed in infected T cell clones, and suggests that H2AK119ub1 is re-established in cells intermittently expressing the provirus.

Knockdown of *USP14*, *OTUD5* or *BAP1* also resulted in the reduction of *shbz* transcription. Paired with the observed correlation between sense and antisense transcription, our results add to previously reported evidence of overlap between the factors regulating expression from the two proviral LTRs. High sense-strand transcript counts correlate with increased transcriptional bursting from the antisense promoter [10], and the presence of Tax itself was found to increase expression from the 3′ LTR [91,92].

Our results differ in one respect from earlier work reporting an increase in *shbz* levels in CD8-depleted PBMCs from HTLV-1-infected individuals following 17-hour treatment with 50 μM PR-619 [26]. It is possible that this discrepancy is due to secondary or additional effects associated with the broad-spectrum DUB inhibition by PR-619.

Whilst *BAP1* is a component of the PR-DUB complex, and its knockdown is known to induce the spread of H2AK119ub1 [93], *OTUD5* and *USP14* may influence proviral transcription through distinct mechanisms. USP14 is recognised as a proteasome-associated DUB, its inhibition increasing proteasome activity [94]. Proteasomal function is key for the processing of both p105 and p100 during NF-$\kappa$B activation [95] and USP14 inhibition or knockdown has specifically been implicated in the suppression of NF-$\kappa$B signaling [96,97]. OTUD5 has been associated with stabilisation of chromatin modifiers, increasing chromatin accessibility during embryonic development [98]. However, knockdown of either *USP14*, *OTUD5* or *BAP1* resulted in a reduction of the other DUBs tested, including *USP16*. All three enzymes were differentially expressed in infected cells, showing increased expression in the 0–24 hour interval relative to uninfected controls. We therefore suggest that this joint regulation is caused by the disruption of proviral transcription following knockdown of any one of these proteins.

The data presented here demonstrate key changes in host gene expression following proviral reactivation in primary PBMCs from HTLV-1 carriers. These changes are contrasted to the transcriptomes of uninfected control samples faced with identical stimuli. The results demonstrate the induction of genes involved in the increased replication of infected cells in response to stimulation, the induction of genes involved in horizontal transmission, and chronic activation of core signalling pathways NF-$\kappa$B and mTORC1 in infected cells. The relationship between PcG gene expression and HTLV reactivation was investigated, showing differential expression of multiple components of PRC1 and PRC2 in infected samples, which might contribute to proviral latency. Finally, DUBs *USP14*, *OTUD5* and *BAP1* were found to influence proviral reactivation, potentially through distinct mechanisms. Whilst a broad range of processes have been investigated here, this analysis represents only a small subset of the total analyses which could be performed using this dataset. To facilitate its further use and allow for rapid browsing of specific transcript trajectories, a browser-compatible application has been launched.

## Materials & methods

### Ethics statement

Blood samples from people living with HTLV-1 (Table 1) were donated by patients attending the Communicable Diseases Research Tissue Bank, approved by the UK National Research Ethics Service (Communicable Diseases Research Tissue Bank ref.: 20/SC/0226). Written informed consent was obtained from all donors, in accordance with the Declaration of Helsinki. Uninfected volunteers (Table 1) consented to sample collection and genomic analysis.

### Peripheral blood mononuclear cell isolation

Peripheral blood mononuclear cells (PBMCs) were isolated from whole blood samples by density gradient centrifugation, using Histopaque-1077 Hybri-Max (Merck, Cat. No. H8889).

**Table 1. HTLV-1$^+$ subject and uninfected volunteer samples used.**

| Sample ID | Sex | Age | HTLV-1 status | Disease status | Experiment |
|-----------|-----|-----|---------------|----------------|------------|
| HES | F | 77 | Infected | Asymptomatic | RNA-seq |
| HQL | F | 63 | Infected | Asymptomatic | RNA-seq/smFISH |
| TCL | F | 65 | Infected | HAM/TSP | RNA-seq |
| TCQ | M | 79 | Infected | HAM/TSP | RNA-seq |
| TDK | M | 74 | Infected | HAM/TSP | RNA-seq |
| TDX | F | 65 | Infected | HAM/TSP | RNA-seq |
| V1 | F | 33 | Uninfected | NA | RNA-seq |
| V3 | M | 24 | Uninfected | NA | RNA-seq and ddPCR timecourse |
| V4 | F | 25 | Uninfected | NA | RNA-seq |
| TCQ | M | 77 | Infected | HAM/TSP | ddPCR timecourse |
| TDT | F | 47 | Infected | HAM/TSP | ddPCR timecourse |
| TET | F | 56 | Infected | HAM/TSP | ddPCR timecourse |
| HHL | F | 34 | Infected | Asymptomatic | smFISH |

Cells were then frozen in FCS (ThermoFisher Scientific, Cat. No. 10500064) 10% DMSO (v/v) and stored at -150°C until use.

## Fluorescence-activated cell sorting of infected and uninfected samples

Frozen PBMC samples were thawed by the addition of room-temperature PBS, and cells were subsequently washed in PBS. Cell counts and viability were measured using the LUNA-FL dual fluorescence cell counter (Logos Biosystems, Cat. No. L20001). CD4$^+$ T cells were enriched using the EasySep Human CD4$^+$ T Cell Enrichment Kit (STEMCELL Technologies, Cat. No. 19052) according to the manufacturer's instructions. Cells were then stained using a viability dye (ThermoFisher Scientific, Cat. No. L10119) and subsequently with biotinylated $\alpha$-CADM1 antibody (MBL, Cat. No. CM004-6) and streptavidin-conjugated BV421 secondary antibody (BioLegend, Cat. No. 405226). Cells were resuspended in 4°C PBS 2% FCS before cell sorting.

Cell sorting was performed using a BD FACSAriaIII instrument, with purity defining sort precision. CADM1$^{high}$ infected cells were sorted based on the CADM1 signal intensity of uninfected cells. Uninfected controls were sorted solely based on size, granularity, forward-scatter height to area ratio (singlet gating) and viability. Sorting was performed at 4°C into 5 ml poly-styrene FACS tubes (Corning, Cat. No. 352054) pre-filled with 200 $\mu$l FCS.

## ddPCR for PVL measurement

Droplet digital PCR (ddPCR) was used to quantify proviral load, as described in Katsuya et al. (2019). DNA was extracted from cells at $t_0$, using a DNeasy Blood and Tissue kit (Qiagen, Cat. No. 69504) and quantified using a NanoDrop 1000 spectrophotometer (Thermo Fisher Scientific). Where possible, genomic DNA dilutions of 5 and 2.5 ng/$\mu$L were prepared. Otherwise, gDNA was used at the extracted concentration, and at an additional dilution. Proviral load was calculated as: (HTLV pX copy number) / (Albumin copy number / 2) · 100. Primer and probe sequences are shown in Table 2; cycler programme in Table 3.

## Cell culture

Sorted cells were cultured in 2 ml RPMI 5.5 mM glucose (1:1 mixture of Cat. No. 11875093 and Cat. No. 11879020 [Thermo-Fisher Scientific]) supplemented with 20% FCS (v/v), Penicillin/Streptomycin (100 U/ml and 100 $\mu$g/ml, Thermo Fisher Scientific, Cat. No. 15140122) and 10

**Table 2. ddPCR primer and probe sequences [33].**

| ID | Sequence (5′ to 3′) |
|---|---|
| tax FWD | CGGATACCCAGTCTACGTGTT |
| tax REV | CAGTAGGGCGTGACGATGTA |
| tax PROBE | /56-FAM/CTGTGTACA/ZEN/AGGCGACTGCC/3IABkFQ/ |
| alb FWD | TGCATGAGAAAACGCCAGTAA |
| alb REV | ATGGTCGCCTGTTCACCAA |
| alb PROBE | /5HEX/TGACAGAGT/ZEN/CACCAAATGCTGCACAGAA/3lABkFQ/ |

$\mu$M raltegravir (Selleck Chemicals, MK-0518), in polystyrene round bottom 5 ml tubes (Corning, Cat. No. 352054). Cells were harvested at 0, 24, 48, 96, 120 and 144 hours, for viability and density measurements and RNA extraction. Each timepoint was cultured in a separate tube.

## RNA extraction

Cultured cells were washed once in 10 ml PBS before RNA extraction using the RNeasy Plus Micro kit (Qiagen, Cat. No. 74034). RNA was eluted in 30 $\mu$l RNase free $H_2O$ and stored at -80˚C until use.

## qRT-PCR

3 $\mu$l (10%) of extracted RNA was used for preliminary analysis of proviral transcription using qRT-PCR. cDNA was generated using the Transcriptor First Strand cDNA Synthesis Kit (Roche, Cat. No. 04379012001), following the manufacturer's instructions. Random nucleotide hexamers were used to prime the reverse transcription reaction.

qPCR of cDNA was performed in 384-well plates (Thermo Fisher Scientific, Cat. No. 4309849), using Fast SYBR Green Master Mix reagent (Thermo Fisher Scientific, Cat. No. 4385612) for amplification detection. A ViiA 7 Real-Time PCR System (Thermo Fisher Scientific) was used to amplify and quantify cDNA. The relevant cycling protocol is shown in Table 4, and the primer sequences used in Table 5.

Prior to relative target quantification, background signal threshold and primer efficiency values ($E$) were calculated for each primer target using LinRegPCR [99]. Target RNA quantity ($R_0$) was calculated as $Threshold/E^{C_q}$, where $C_q$ is the quantification cycle where a reaction's fluorescent signal intensity crosses the background threshold. $R_0$ values for each target transcript were then normalised relative to 18S rRNA levels.

## Sequencing

Prior to sequencing, extracted RNA quality and concentration were measured using the Bioanalyser RNA Pico kit (Agilent, Cat. No. 5067–1513). Library preparation and sequencing were performed by the Oxford Genomics Centre (https://www.well.ox.ac.uk/ogc/). PolyA-enriched

**Table 3. Thermal cycler programme used for ddPCR.**

| PCR step | Temperature (˚C) | Duration (min) | # cycles |
|---|---|---|---|
| Initial activation | 95 | 10 | 1 |
| Denaturation | 94 | 0.5 | 40 |
| Annealing/extension | 58 | 1 | 40 |
| Enzyme deactivation | 98 | 10 | 1 |
| Hold | 4 | $\infty$ | 1 |

**Table 4. Thermal cycler protocol for qPCR.**

| Step | Temperature (°C) | Duration |
|---|---|---|
| Hold 1 | 50 | 2' |
| Hold 2 | 95 | 10' |
| Melt (45×) | 95 | 15" |
| Elongation (45×) | 60 | 1' |
| Melt curve 1 | 95 | 15" |
| Melt curve 2 | 60 | 1' |
| Dissociation | 95 | 15" |

cDNA libraries were prepared. 150 bp paired-end fragments were sequenced using the Nova-Seq6000 sequencing platform (Illumina). Reads for each sample were collected over two individual sequencing runs.

## Alignment

**STAR alignment.** An alignment to visualise reads aligning to individual parts of the HTLV provirus was performed using STAR v2.7.9a [36]. Reads were aligned to the GRCh38 assembly of the human genome, with HTLV-1 genome sequence AB513134 [100] appended as an additional chromosome. A GTF file containing genomic feature information for the human genome (ref. GRCh38) was obtained from the Ensembl database and the coordinates for HTLV-1, excluding the LTRs, were appended. Reads cannot be unambiguously mapped to either LTR due to their near-identical sequence, hence these regions were excluded.

**Kallisto pseudo-alignment.** Transcript counts were estimated using Kallisto version 0.46.2 [37]. The transcript index was generated from cDNA sequences for GRCh38 obtained from Ensembl. The entire length of the provirus (AB513134), excluding the LTRs, in the sense and antisense orientations were used as transcript sequences to quantify HTLV-1 sense and antisense transcripts respectively.

## Differential expression analysis

Transcript differential expression analysis was performed on Kallisto transcript counts. Linear models for comparison of sample transcript count matrices were generated using the DESeq2 package [38] for R [101]. Models were generated iteratively: first incorporating differences between time in culture, then adding the effect of HTLV-1 infection and, finally, adding the interaction term between infection status and time in culture. Likelihood Ratio Tests were used to quantify the number of genes identified as significant by more complex models.

To obtain sets genes of genes differentially expressed between sequential timepoints in infected cells, DESeq2 models were generated using the same data, switching the intercept to allow for specific pairwise comparisons. Global $p$-value correction was then used to correct for

**Table 5. Primers used for qRT-PCR [29].**

| ID | Sequence FWD (5′ to 3′) | Sequence REV (5′ to 3′) |
|---|---|---|
| *tax/rex* | CCGGCGCTGCTCTCATCCCGGT | GGCCGAACATAGTCCCCCAGAG |
| *gag* | CAGAGGAAGATGCCCTCCTATT | GTCAACCTGGGCTTTAATTACG |
| *pX* | CTCCTTCCGTTCCACTCAAC | GTGGTAGGCCTTGGTTTGAA |
| *18S rRNA* | GTAACCCGTTGAACCCCATT | CCATCCAATCGGTAGTAGCG |

the multiple pairwise comparisons performed using the Benjamini-Hochberg false discovery rate (BH-FDR) [60].

### Identification of sense-strand correlated genes

Sense-strand expression correlates were obtained by performing pairwise Spearman rank correlation tests between sense-strand transcript counts and all other identified transcripts. $p$-value correction for multiple hypothesis testing was subsequently performed using BH-FDR.

### Over-representation analyses

Over-representation analyses (ORA) were performed using the clusterProfiler package for R [39]. Ensembl gene identifiers were converted to Entrez IDs using biomaRt [102,103]. The clusterProfiler::enrichGO function was used in conjunction with the org.Hs.eg.db database [104] to perform ORA against the biological process clade of the Gene Ontology (GO) [40,41]. A cutoff value of 0.05 for FDR-corrected $p$-values was set for significant results. A gene background of all the genes quantified in the Kallisto pseudoalignment was used. For the ORA performed in Fig 3, GO terms in levels < 5 and > 10 were removed using the clusterProfiler::dropGO function. Remaining redundant terms were removed using the clusterProfiler::simplify function with a 50% overlap threshold, retaining the term with the lowest adjusted $p$-value.

### Horizontal infection culture

For the ddPCR measurements of proviral copies over time presented in S7A and S7B Fig, patients' samples (Table 1) were thawed from -150˚C storage, as described above, enriched for CD4+ T cells as described above, and subsequently cultured at 37˚C, 5% $CO_2$ in 2 ml RPMI 5.5 mM glucose supplemented with 20% FCS (v/v), Penicillin/Streptomycin (100 U/ml and 100 $\mu$g/ml), in 5 ml polystyrene FACS tubes without the addition of raltegravir. DNA was extracted as for other ddPCR samples, and quantification was performed following the protocol described by Katsuya et al. [33].

Samples used in the smFISH experiments presented in S7C and S7D Fig (Table 1) were enriched for CD4+ T cells, stained and sorted for high CADM1 expression, and cultured, all as described above, in the cell culture section for the samples used for RNA-seq.

### Coverslip preparation

Twelve-millimeter diameter circular No. 1 coverslips (Scientific Laboratory Supplies, Cat. No. MIC3304) were washed in 100% EtOH and then placed inside individual wells of 12-well plates (Corning, Cat. No. 353043) to dry. After drying, coverslips were covered with 100 $\mu$l of 0.1% (w/v) poly-L-lysine solution (Merck, Cat. No. P8920) for 10 minutes at room temperature. Coverslips were then washed twice with 1 ml $H_2O$ and left to dry for $\approx$ 30 minutes. Cells were washed $2 \times$ in PBS and layered onto coverslips, in PBS ($\approx$ 1250 cells/$\mu$l), and left to sediment for 10 minutes, room temperature. Two 1 ml PBS washes were then performed to remove unadhered cells. Cells were fixed for 1 minute in 1 ml PBS 2% formaldehyde (Thermo-Fisher Scientific, Cat. No. 28906) before being fixed for a further 14 minutes in fresh PBS 2% formaldehyde. Following two PBS washes, cells were permeabilised with 4˚C 70% EtOH and stored at -20 ˚C until imaging.

### Fluorescent probe hybridisation

Coverslips were removed from EtOH and excess liquid removed by blotting on filter paper. Coverslips were subsequently submerged in wash buffer A (10% [v/v] de-ionised formamide,

20% [v/v] Stellaris wash buffer A [LGC Biosearch Technologies, Cat. No. SMF-WA1-60], 70% $H_2O$) for 10–15 minutes. The Stellaris RNA probe mixture (S3 Table) was prepared by adding 1:100 of the relevant probes (final concentration 40 $\mu$M) to hybridisation buffer (10% [v/v] de-ionised formamide, 90% [v/v] Stellaris hybridisation buffer [Cat. No. SMF-HB1-10]). Probe mixture was pipetted onto a hybridisation cover (Thermo-Fisher Scientific, Cat. No. H18200) resting on a glass microscope slide (Thermo-Fisher Scientific, Cat. No. J2800AMNZ). Coverslips were removed from wash buffer A, blotted and placed onto the probe mix, cells facing down. Coverslips were then covered in rubber cement (Marabu, Cat. No. 29010017000), to prevent evaporation, and incubated overnight at 37˚C, protected from light. Following hybridisation, coverslips were incubated in wash buffer A for 30 minutes at 37˚C, 60 RPM. Coverslips were then transferred into fresh wash buffer A supplemented with 5 ng/ml DAPI and incubated for 30 minutes at 37˚C, 60 RPM. After 5-minute incubation in Stellaris wash buffer B (Cat. No. SMF-WB1-20) at room temperature, cells were blotted and mounted onto glass slides in Vectashield antifade mounting medium (Vector Laboratories, Cat. No. H-1000-10). Coverslip edges were sealed with transparent nail polish and left to set for 10 minutes prior to imaging.

### smFISH probes

Novel Stellaris smFISH probes targeting proviral transcripts were designed using the manufacturer's browser-based design tools (https://www.biosearchtech.com/support/tools/design-software/stellaris-probe-designer) (S3 Table).

### Microscopy

Imaging was performed on an Olympus IX70 inverted fluorescence microscope (Olympus Corporation) fitted with an ORCA-Flash 4.0 V2 digital CMOS camera (Hamamatsu Photonics), using the 100× objective. Micromanager 2.0 software [105] was used to coordinate the acquisition of z-stacks in four channels (Table 6). 54 z-stack slices for each field of view were obtained at 300 nm intervals.

### Image processing

Images were first corrected for uneven illumination using channel-specific illumination functions as used by Billman, Rueda, and Bangham [10] and Miura et al. [11]. Excess image slices (≤4) at either end of the z-stack were removed in parallel, based on the focal centre of the stack, where the standard deviation of intensity is greatest.

### Clonal infected T cell culture

Clones 11.50, 11.63 and TBX4B [34] (Table 7) were cultured in RPMI-1640 supplemented with Penicillin/Streptomycin (100 U/ml and 100 $\mu$g/ml), L-Glutamine and 20% FCS (v/v), 100 U/ml IL-2 (Miltenyi Biotec, Cat. No. 130-097-745) and 10 $\mu$M raltegravir (Selleck Chemicals,

**Table 6. Channels used in acquisition of smFISH images.**

| Channel | Excitation $\lambda$ (nm) | Emission $\lambda$ (nm) | Exposure time (ms) |
|---|---|---|---|
| DAPI | 390/18 | 457/20 | 100 |
| Autofluorescence | 475/28 | 514/30 | 500 |
| Quasar 570 | 575/25 | 632/60 | 500 |
| Quasar 670 | 632/22 | 692/40 | 500 |

**Table 7. Details of clonal HTLV-1-infected T cells used.**

| Clone | Patient code | Proviral integration site (GRCh38) |
|-------|-------------|-----------------------------------|
| 11.50 | TBW | Chr 19: 27,791,679 |
| 11.63 | TBW | Chr 19: 33,338,642 |
| TBX4B | TBX | Chr 22: 43,927,318 |

MK-0518) at 5% $CO_2$, 37°C. Transduced lines were maintained in 1 $\mu$g/ml puromycin (Thermo-Fisher Scientific, Cat. No. A1113803). Replacement of half the culture media with fresh media was performed 2–3 times per week.

## RNA interference

Transcript knockdown was performed using Dharmacon Accell SMARTpool siRNA (Horizon Discovery) which incorporates four siRNA sequences for a target to knock down gene expression. $2 \cdot 10^5$ cells were resuspended in 200 $\mu$l Accell delivery media supplemented with 100 U/ml IL-2, 10 $\mu$M raltegravir and 1 $\mu$M of the relevant SMARTpool siRNA in a V-bottom 96-well plate. Cells were incubated at 37°C, 5% $CO_2$ for 72 hours before lysis and RNA extraction using the RNeasy Micro Plus extraction kit (Qiagen, Cat. No. 74034). A further technical replicate was resuspended in RPMI supplemented with 20% FCS, 100 U/ml IL-2 and 10 $\mu$M raltegravir, cultured for a further 24 hours before lysis and RNA extraction. Primer sequences for deubiquitinase quantification are shown in Table 8.

siRNA sequence alignment was performed using Clustal Omega [106,107] and visualised using Jalview v2.11.2.2 [108].

## Motif search

To identify partial matches to E- and T-box motifs in the HTLV-1 sequence (AB513134), FIMO [109] was used to identify matches at a significance threshold of 0.001. E- and T-box motifs used (MA0058.3 and MA0690.2 respectively) were obtained from JASPAR [110].

**Table 8. Primers used to examine candidate deubiquitinase expression levels.** Primers were designed against the "Ensembl Canonical" transcript variant for each gene. Annealing sites matching exon-exon splicing points were selected.

| ID | FWD (5′ to 3′) | REV (5′ to 3′) |
|----|---------------|---------------|
| ATXN3 | CCATCTTCCACGAGAAACAAGA | GTAACTCCTCCTTCTGCCATTC |
| OTUD5 | GCCGACTACTTCTCCAACTATG | GATGGGTTCTGTGCTGTACT |
| BAP1 | AAATACTCACCCAAGGAGCTG | CTGAGCCAGCATGGAGATAAA |
| USP14 | GTGGATTGACAAACCTTGGTAAC | CAAATCTCTAAGGGCTGCAGTA |
| USP5 | GGGAAACAGTATGTGGAGAGAC | TCCGCCTTCAACACCAATAG |
| UCHL5 | GCAGTAAGGCCTGTCATAGAA | GATCTGTATCCATGGGTTCCTC |
| UCHL3 | CCTGAAGAACGAGCCAGATAC | TTCCGCCCATCTAATTCATAGAG |
| USP22 | GGCCACTACACCAGCTTTAT | TGATAGAACAGCAAGTACCCTTC |
| USP9X | GCTGAACGAATGGCAGAATG | CCTGTGCTGCCCAGATATTA |
| USP8 | TACGATGGCAGGTGGAAAC | GCCCACCGTAGTGATTTGA |
| USP16 | CCTCACTGTCTGGTTCTTAGTTT | CATTATCTTTCTCTGCTGGCTTTG |
| USP47 | GCATTTGCTAGTGTGGAAGAAG | AGCTGTAAGGTCAGCAGATAAG |
| USP19 | GAGGAGGAAGAGGAAGAGAAGA | CAAAGGAGCGGTCATGGAA |
| VCPIP1 | AACCATGGGTATGGCTGATG | GAACCTGACTTACAGCCTCTTT |

## Statistical analyses and plotting

Statistical analyses were performed using R statistical computing software [101]. Data were plotted using ggplot2 [111], with the exception of flow cytometry scatter and contour plots, which were generated using FlowJo v10.8.1 (BD Life Sciences).

## Supporting information

**S1 Fig. Proviral transcript expression following reactivation.** HTLV-1 sense-strand RNA trajectories during ex vivo culture, obtained using qRT-PCR."gag" amplicons correspond to proviral region 2,017–2,203, whilst"pX" corresponds to region 8,000–8,161. tax/rex amplicons straddle the second exon junction. Coordinates presented for HTLV-1 sequence AB513134. Dark points and dotted lines represent mean ± SEM.
(TIF)

**S2 Fig. Aberrant reads mapping to uninfected sample at single timepoint.** Values represent number reads mapped to HTLV sense or antisense strands normalised to the total number of reads mapped to hg38, with AB513134 appended as an additional chromosome, by STAR.
(TIF)

**S3 Fig. Antisense-strand transcription correlates with sense-strand reactivation.** (A) DESeq2-normalised counts for sense and antisense strand transcripts normalised to their maximum values. (B) DESeq2-normalised counts of reads aligned to the proviral sense or antisense-strand exons. Linear model fit with 95% confidence interval shown. Statistics shown from Spearman's rank correlation test.
(TIF)

**S4 Fig. Age discrepancy between infected and uninfected samples.** Uninfected controls were not age-matched with infected patients. p-value shown from Wilcoxon test.
(TIF)

**S5 Fig. Clustering of PBMC samples relative to fluorescent timer protein populations described in Kiik et al. [22].** PCA analysis performed on VST-normalised data, subsequently z-scaled to correct for read count discrepancies between datasets.
(TIF)

**S6 Fig. Cells progress through the cell cycle following the peak in sense-strand expression.** Mean trajectories of genes with peak expression levels at distinct cell cycle stages. Shaded regions represent ± SEM.
(TIF)

**S7 Fig. Changes in genes associated with horizontal infection over expression cycle.** (A) Left: ddPCR PVL measurements for three infected samples and one uninfected control (V3). Right: Repeat measurement of 96-hour infected-cell samples, together with 11.50 positive control for 100% PVL. Error bars represent ± SD of two technical replicates consisting of sample dilutions. RT refers to reverse-transcribed. (B) Final timepoint from panel A, with 11.50 positive control for 100% PVL included. Error bars ±SEM. RT refers to reverse-transcribed. (C) Above: Schematic of HTLV-1 provirus and relative positioning of smFISH probes. Below: Example of cell showing clustering of unspliced proviral RNA near cytoplasmic periphery. Scale bar 5 μm. (D) Quantification of cells with visible clusters of unspliced RNA. (E) Trajectories of genes reported to influence horizontal infection. Asterisks indicate genes which change significantly over time and relative to uninfected cells. Summary lines and shaded areas

represent mean ± SEM.
(TIF)

**S8 Fig. Distribution of E- and T-box motifs in HTLV-1.** (A) Sequence logos of E- and T-box motifs obtained from JASPAR [110]. (B) Sequence logos of significantly matching (p < 0.001) sequences in HTLV. Generated using ggseqlogo [113]. (C) Distribution of sequences with significant similarity to E-box and T-box motifs along provirus. Grey boxes represent LTRs.
(TIF)

**S9 Fig. Alignment of siRNA sequences used for DUB RNAi reveals no overt similarity between co-affected genes.** Sequences of siRNA fragments used to knockdown DUB transcripts, aligned using Clustal Omega [107].
(TIF)

**S10 Fig. FOS already upregulated at time of 0-hour measurement.** Lines and shaded areas represent mean ± SEM.
(TIF)

**S1 Table. Ranking of histone-associated GO: BP terms from ORA against sense-strand correlates.**
(XLSX)

**S2 Table. Genes included in histone associated terms from ORA of sense-strand correlated genes.**
(XLSX)

**S3 Table. smFISH probe sequences.**
(XLSX)

## Acknowledgments

We thank Parisa Amjadi from the CL3 Cell Sorting Facility at The Centre for Immunology and Vaccinology at Imperial College London. We thank Theodosios Kyriakou and the Oxford Genomics Centre for library preparation and RNA Sequencing. We thank Rob Klose (University of Oxford) for a fruitful discussion regarding Polycomb group proteins, Helen Kiik and Anat Melamed (Imperial College London) for bioinformatics advice. Aileen Rowan (Imperial College London) for additional data on the samples used in this study. We thank the Imperial College Research Computing Service for use of the high-performance computing cluster (https://doi.org/10.14469/hpc/2232).

## Author Contributions

**Conceptualization:** Aris E. N. Aristodemou.

**Data curation:** Aris E. N. Aristodemou.

**Formal analysis:** Aris E. N. Aristodemou.

**Funding acquisition:** David S. Rueda, Charles R. M. Bangham.

**Investigation:** Aris E. N. Aristodemou.

**Methodology:** Aris E. N. Aristodemou.

**Project administration:** Aris E. N. Aristodemou, Charles R. M. Bangham.

**Resources:** Aris E. N. Aristodemou, David S. Rueda, Graham P. Taylor.

**Software:** Aris E. N. Aristodemou.

**Supervision:** Charles R. M. Bangham.

**Validation:** Aris E. N. Aristodemou.

**Visualization:** Aris E. N. Aristodemou.

**Writing – original draft:** Aris E. N. Aristodemou, Charles R. M. Bangham.

**Writing – review & editing:** Aris E. N. Aristodemou, David S. Rueda, Graham P. Taylor, Charles R. M. Bangham.

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
