## [Decision Letter · Decision Letter 0]

20 Feb 2023

Dear Dr. Bagham,

Thank you very much for submitting your manuscript "The transcriptome of HTLV-1-infected primary cells following reactivation reveals changes to host gene expression central to the proviral life cycle" for consideration at PLOS Pathogens. As with all papers reviewed by the journal, your manuscript was reviewed by members of the editorial board and by several independent reviewers. In light of the reviews (below this email), we would like to invite the resubmission of a significantly-revised version that takes into account the reviewers' comments. There was a bit of divergence of opinion of the three reviewers, but I do do agree with many of the points raised by reviewer #2.

We cannot make any decision about publication until we have seen the revised manuscript and your response to the reviewers' comments. Your revised manuscript is also likely to be sent to reviewers for further evaluation.

Sincerely,

Patrick L Green

Guest Editor

PLOS Pathogens

Susan Ross

Section Editor

PLOS Pathogens

Kasturi Haldar

Editor-in-Chief

PLOS Pathogens

orcid.org/0000-0001-5065-158X

Michael Malim

Editor-in-Chief

PLOS Pathogens

orcid.org/0000-0002-7699-2064

Reviewer's Responses to Questions

**Part I - Summary**

Reviewer #1: In this manuscript, Aristodemou et al. analyzed global host gene expression following reactivation of HTLV-1 proviral genes in ex-vivo cultures of primary infected CD4+ T-cells. This is a comprehensive study that compares gene expression from six HTLV-1 positive patients with that from three non-infected patients over a six-day period. Interestingly, the authors observed great homogeneity in gene expression within sample groups (infected or not). There are several important findings in this study, with the main ones being that genes encoding certain deubiquitinases are involved in viral re-expression while genes encoding proteins of the polycomb repressive complex (PRC1 and PRC2) are linked to viral repression. The information obtained from this study adds to knowledge of how HTLV-1 orchestrates cellular events to promote viral expansion. In addition, the authors have made publicly available the large quantity of data obtained for this study into a user-friendly website. The comments below would help clarify some aspects of the manuscript.

Reviewer #2: Aristodemou et al performed transcriptome analysis of host and viral genes in CADM-1+ CD4+ T cells sorted from HTLV-1-infected and uninfected subjects at pre- and post- ex vivo culture and demonstrated that host gene expression patterns in the infected cells were changed and different from those in uninfected cells at pre-culture as well as at each time point of culture depending on the period of proviral reactivation or horizontal viral transmission. In addition to characteristic gene expressions related to chronic activation such as NF-kB and mTORC1 pathway, the authors demonstrated that the Polycomb group of epigenetic genes were differentially expressed in the infected cells compared to the uninfected cells during the viral reactivation. Moreover, they identified some deubiquitinases can promote proviral transcription ex vivo culture of the infected cells. It is of interest to determine and understand the complex mechanism of the regulation in HTLV-1 latency and reactivation using the primary infected cells isolated from HTLV-1-infected subjects.

The authors showed and well discussed the results with multiple data analysis and the supportive experiments, but the manuscript is not well organized due to the lack of or unclear information. For examples, there were three tables for characteristics of samples and the method sections about sample collection and sorting in duplicate in the manuscript. The organization and explanation of human biological samples and the associated experimental methods are also required.

Reviewer #3: Aristodemous et al. analyze the temporal changes in viral and host cell transcription during HTLV-1 reactivation in CD4+ cells freshly isolated from patients’ peripheral blood over six-day cultures.

Among the different changes observed, the Authors focus their attention on the expression of the Polycomb epigenetic modifiers, which may form a negative-feedback mechanism allowing the virus to return to a latent state following activation. This study also identified a three deubiquitinases that increase viral expression.

This is an excellent study that applies state-of the-art techniques to tackle some key unanswered questions on the regulation of HTLV-1 gene expression and its connection with the pattern of host gene expression. The experimental layout is sound and the data presented are solid.

The results confirm and extend previous work from different labs (including Prof. Bangham’s) that analyzed the kinetics of expression of the different viral genes and their impact on the host cell transcriptome.

**Part II – Major Issues: Key Experiments Required for Acceptance**

Reviewer #1: 1) The finding that both sense and anti-sense transcription are activated early, and correlate is very interesting and could be incorporated into the main manuscript. This observation is important because a role of the antisense transcript/protein in viral replication has been controversial given an earlier study from Belrose et al. (2011, Blood), which found an inverse correlation between sense and anti-sense transcripts. Along these lines, the authors should discuss the different conclusion of this study to that of one of their former studies (Kiik et al., 2022, Plos Pathogens). In the previous publication they suggested that Tax was unlikely to directly regulate HBZ expression.

2) To explain the correlation above, the authors discuss a possibility that Tax regulates anti-sense transcription. However, if this mechanism occurred, one might expect a delay between sense (tax/rex) and anti-sense transcription? Could the same host factors regulating sense transcription also regulate anti-sense transcription? Specifically, is there a correlation between the expression of these factors and anti-sense transcription?

3) As done for the deubiquitinases, it would be interesting to use siRNA targeting certain polycomb genes to determine whether they are linked to a reduction in sense (and anti-sense) transcription

Reviewer #2: Main points:

1. It is not clear how to determine the infection status in HTLV-1-infected subjects and the disease status in asymptomatic carriers and HAM/TSP patients. In addition, it was shown that one of asymptomatic carriers showed HTLV-1 negative in the sorted CADM1+CD4+ T cells by ddPCR. Please describe the criteria of HTLV-1-infected subjects.

2. The authors did not discuss the difference of the infected CD4+ T cell samples on each result whether the infected samples were obtained from asymptomatic carriers or HAM/TSP patients. It is important if there was any difference or no difference by disease status in the study. It would be more informative if the authors can provide HTLV-1 PVL in PBMC of each HTLV-1 subjects and disease progression/activity of HAM/TSP patients used in the study.

3. In the manuscript, asymptomatic carriers (also sometime uninfected samples) are described as “HTLV-1 patient” in the text, figures 4, 5 and 6, and table 6 and 8. Please carefully describe the group of HTLV-1-infected and uninfected subjects.

4. The authors mentioned that the negative control sample V4 t24 had a contamination of HTLV and was excluded for further analysis. In the results, total number of uninfected controls were different at each time point (2 or 3 controls). Since the authors also described the importance of uninfected controls for distinguishing the pathways and factors related to HTLV-1 proviral reactivation, it would be helpful to explain whether the removal of V4 t24 did or did not affect the analysis of differential gene expression by time course.

5. Lanes 127 and 460-462, Fig S4: The authors mentioned that the substantial difference in mean age between the infected and uninfected subjects might contribute to their results. Is there any association of CADM1 expression on CD4+ T cells with aging? Is there any differential gene expression pattern strongly associated with aging in CADM1+CD4+ T cells compared to the uninfected cells?

6. Lane 133-143: In the section, the authors described that the current study data were compared with the data set obtained by (22). But it is not clear what the data set were obtained from the study ((22) Kiik et al 2022). If the authors used the transcriptome data used in the study (Kiik et al 2022), it is better to explain if the T cell clones used in the study (described as CD4+CD25+CCR4+ T cells, maintained with human IL-2; Kiik et al 2022) were comparable with CADM1+CD4+ T cells which were used in the current study. In addition, it is also not clear in the figure legend of Fig S5 which is associated with the result section.

7. Cell culture: It is described that the cells were cultured in RPMI medium with 10uM of raltegravir (lane 592 and 671), but not in the other methods (lane 666 and 697). If there is any specific reason to culture the cells with or without the integrase inhibitor, it is better to include clear description of the culture condition in each experiment.

8. Cell culture: How long raltegravir is effective on prevention of secondary HTLV-1 infection in the culture? Please describe if the sorted cells were cultured with or without any additional supplement up to 144 hours.

9. Lane 188-203: In the section, the authors described about horizontal viral infection during ex vivo culture. According to the methods (lane 666), the culture condition seems to be different from the culture condition for the sorted cells (lane 592) and patient’s samples (table 6) are also different from the ones used for the sorted cells (table 1). How were the results evaluated for the possibility of horizontal infection in the ex vivo culture of the sorted cells?

10. Lane 204-213: In the section, the additional samples from HTLV-1 infected asymptomatic carriers were used for the experiment (table 8) which are different from the ones used for the sorted cells (table 1). The method was not clearly described in the associated section of materials and methods (lane 697). Therefore, it is not clear if the results were comparable to the data generated from the ex vivo culture of the sorted cells and how the authors evaluated the possibility of horizontal infection in the ex vivo culture of the sorted cells by the results.

11. It is important to explain the use of human biological specimens in research. Please make a table listing all the human subjects used in the study, describe each method by experiment clearly and refer the table correctly in the manuscript.

12. Three clonal infected T cell lines were used for the experiment of RNA interference targeting DUBs. Did the clonal infected T cell lines represent similar DUB trajectories with the sorted primary HTLV-1-infected cells compared to uninfected cells? The authors also described that serum was reintroduced at 72 hours, following t72 RNA sample collection (lane 383). What does “Serum” mean?

13. Using siRNA knockdown of each target, the authors concluded three DUBs (USP14, OTUD5 and BAP1) were associated with proviral reactivation. However, the expression of BAP1 was stable or gradually downregulated in the infected cells after cell culture for 24 hours compared to the uninfected cells, while USP14 and OTUD5 were significantly upregulated. In the uninfected cells, the change and pattern of these three gene expressions were different each other and from those in the infected cells. Are these three and/or the other targets of PR-619 directly or specifically involved in HTLV-1 reactivation? In addition, it was very low RNA expressions of HTLV-1 and host transcripts detected in the infected T cell clones at both 72- and 96-hours following siRNA knockdown. Are there any significant differences of RNA expression by siRNA knockdown of each target?

14. Lane 541: A link of a private website should not be included in discussion.

Reviewer #3: (No Response)

**Part III – Minor Issues: Editorial and Data Presentation Modifications**

Reviewer #1: - There is a need to clarify the horizontal transmission (figure S7). There appears to be a combination of both new infection and superinfection events occurring, which should be better explained. Technical details are missing (how many cells used in what volumes). Were the control 11.50 cells also subjected to the same culture conditions, and why did they not also show evidence of superinfection? The apparent level of superinfection was surprising and weakens the notion that LFA-1 is downregulated to prevent this event.

- Would the infection process be expected to have a delay between the formation of unspliced proviral RNA clusters and the increase in RT proviral copies?

- Author names are missing lane 585 and 670.

Reviewer #2: Minor points:

1. Materials and Methods (Lanes 572, 704 and 707): The authors used the abbreviation of room temperature as “RT”. However, “RT” was also used in the manuscript such as “RT-PCR” (lane 603) and “RT proviral copies” (Fig S7). Please clarify and distinguish the abbreviation.

2. Materials and Methods (Lane 583): Please describe which instrument was used for the sorting and how was the purity of the sorted cells.

3. Materials and Methods (Lanes 585 and 670): Please remove the words “(authors?)”.

4. Materials and Methods: There are multiple tables which were not referred in the manuscript.

5. The authors used the same asterisks (***) representing a significant difference in figures 4, 5 and 6, but the statistical methods and comparisons used in each analysis are different among the figures although the x-axis (time) and y-axis (DESeq2-normalized counts) are the same on all the graphs. To avoid the confusion, please change the mark in each figure.

6. Lane 320: (Fig. 5E)?

7. Lane 330: (Fig. 5F)?

8. Lanes 335 and 337: (Fig. 5G)?

Reviewer #3: Below are some comments/suggestions to improve the quality of the paper.

The control individuals are much younger than the infected patients. This may introduce a bias especially for the genes that control epigenetic processes, which are known to be affected by age.

Positive selection by CADM1 may introduce biases due to the activation of signal transduction pathways controlled by this receptor. In fact, Tax function itself may be affected by CADM1 (https://doi.org/10.1371/journal.ppat.1004721).

The Authors might want to consider moving Figure S1 to the main manuscript, as it introduces the concept of the study.

Based on the legend to Figure S1, it seems that the ‘pX’ sequences (8,000–8,161) would comprise all sense-strand transcripts. Please clarify. It would have been interesting to see data on reads straddling other splice-junctions besides tax-rex.

Previous studies (e.g. doi:10.1182/blood-2010-11-316463) indicated that the peak of tax/rex mRNA expression may occur as early as 4-8 hours after start of culture. It would have been useful to see at least one time point earlier than 24 hr in this study.

Why was raltegravir added to the cultures?

PLOS authors have the option to publish the peer review history of their article (what does this mean?). If published, this will include your full peer review and any attached files.

Reviewer #1: No

Reviewer #2: No

Reviewer #3: No
---

## [Decision Letter · Decision Letter 1]

19 Jun 2023

Dear Dr Aristodemou,

We are pleased to inform you that your manuscript 'The transcriptome of HTLV-1-infected primary cells following reactivation reveals changes to host gene expression central to the proviral life cycle' has been provisionally accepted for publication in PLOS Pathogens.

Best regards,

Patrick L Green

Guest Editor

PLOS Pathogens

Susan Ross

Section Editor

PLOS Pathogens

Kasturi Haldar

Editor-in-Chief

PLOS Pathogens

orcid.org/0000-0001-5065-158X

Michael Malim

Editor-in-Chief

PLOS Pathogens

orcid.org/0000-0002-7699-2064

Reviewer Comments (if any, and for reference):

Reviewer's Responses to Questions

**Part I - Summary**

Reviewer #1: The authors adequately addressed all my comments.

Reviewer #3: The revised manuscript is acceptable for publication

Reviewer #4: In the manuscript titled, “The transcriptome of HTLV-1-infected primary cells following reactivation reveals changes to host gene expression central to the proviral life cycle”, the authors examined the expression of host and viral genes during HTLV-1 reactivation in cells freshly isolated from infected and uninfected patients’ blood. A major change in Polycomb group of epigenetic modifiers was observed and certain deubiquitinases were found to promote proviral transcription. This study is novel and provides a detailed account of HTLV-1 proviral reactivation in a short window of time. The authors adequately addressed all reviewer concerns.

**Part II – Major Issues: Key Experiments Required for Acceptance**

Reviewer #1: None.

Reviewer #3: n/a

Reviewer #4: (No Response)

**Part III – Minor Issues: Editorial and Data Presentation Modifications**

Reviewer #1: None.

Reviewer #3: n/a

Reviewer #4: 1) Line 56 – space between cells and in vivo.

2) Line 279 – ‘exclusively the in’, please delete ‘the’.

3) Line 350 – ‘We’ should be lowercase.

4) Line 391 – ‘and other and ubiquitin’ should be ‘and other ubiquitin’.

PLOS authors have the option to publish the peer review history of their article (what does this mean?). If published, this will include your full peer review and any attached files.

Reviewer #1: No

Reviewer #3: No

Reviewer #4: No

---

## [Editor Report · Acceptance letter]

24 Jul 2023

Dear Dr Aristodemou,

We are delighted to inform you that your manuscript, "The transcriptome of HTLV-1-infected primary cells following reactivation reveals changes to host gene expression central to the proviral life cycle," has been formally accepted for publication in PLOS Pathogens.

Best regards,

Kasturi Haldar

Editor-in-Chief

PLOS Pathogens

orcid.org/0000-0001-5065-158X

Michael Malim

Editor-in-Chief

PLOS Pathogens

orcid.org/0000-0002-7699-2064